# Correlation Clustering with Active Learning of Pairwise Similarities

**Linus Aronsson**                                                            *linaro@chalmers.se*
*Chalmers University of Technology*

**Morteza Haghir Chehreghani**                          *morteza.chehreghani@chalmers.se*
*Chalmers University of Technology*

Reviewed on OpenReview: *https://openreview.net/forum?id=Ryf1TVCjBz*

## Abstract

Correlation clustering is a well-known unsupervised learning setting that deals with positive and negative pairwise similarities. In this paper, we study the case where the pairwise similarities are not given in advance and must be queried in a cost-efficient way. Thereby, we develop a generic active learning framework for this task that benefits from several advantages, e.g., flexibility in the type of feedback that a user/annotator can provide, adaptation to any correlation clustering algorithm and query strategy, and robustness to noise. In addition, we propose and analyze a number of novel query strategies suited to this setting. We demonstrate the effectiveness of our framework and the proposed query strategies via several experimental studies.

## 1 Introduction

Clustering constitutes an important unsupervised learning task for which several methods have been proposed in different settings. *Correlation clustering* (Bansal et al., 2004; Demaine et al., 2006) is a well-known clustering problem that is particularly useful when both similarity and dissimilarity judgments are available between a set of objects. In this setting, a dissimilarity is represented by a negative relation between the respective pair of objects, and thus correlation clustering deals with clustering of objects where their pairwise similarities can be positive or negative numbers. Correlation clustering has been employed in numerous applications including image segmentation (Kim et al., 2011), spam detection and filtering (Bonchi et al., 2014; Ramachandran et al., 2007), social network analysis (Tang et al., 2016; Bonchi et al., 2012), bioinformatics (Bonchi et al., 2013b), duplicate detection (Hassanzadeh et al., 2009), co-reference identification (McCallum & Wellner, 2004), entity resolution (Getoor & Machanavajjhala, 2012), color naming across languages (Thiel et al., 2019), clustering aggregation (Gionis et al., 2007; Chehreghani & Chehreghani, 2020) and spatiotemporal trajectory analysis (Bonchi et al., 2013b).

This problem was first introduced with pairwise similarities in $\{-1, +1\}$ (Bansal et al., 2004) and then it was extended to arbitrary positive and negative pairwise similarities (Demaine et al., 2006; Charikar et al., 2005). Finding the optimal solution of correlation clustering is NP-hard (and even APX-hard) (Bansal et al., 2004; Demaine et al., 2006). Thereby, several approximate algorithms have been proposed for this problem (e.g., (Bansal et al., 2004; Demaine et al., 2006; Ailon et al., 2008; Charikar et al., 2005; Elsner & Schudy, 2009; Giotis & Guruswami, 2006)) among which the methods based on *local search* perform significantly better in terms of both the quality of clustering and the computational runtime (Thiel et al., 2019; Chehreghani, 2022).

All of the aforementioned methods assume the pairwise similarities are provided in advance. However, as discussed in (Bressan et al., 2019; García-Soriano et al., 2020), computing such pairwise similarities can be computationally demanding, or they might not even be given a priori and must be queried from a costly oracle (e.g., a human expert). For example, as described in (García-Soriano et al., 2020), computing the

interactions between biological entities may require highly trained professionals spending time and costly resources, or, in the entity resolution task, crowd-sourcing the queries about pairwise similarities can involve a monetary cost. Thus, a fundamental question is concerned with *design of a machine learning paradigm that yields a satisfactory correlation clustering solution with a limited number of queries for pairwise similarities between the objects.*

In machine learning, such a question is usually studied in the context of *active learning* where the goal is to obtain the most informative data given a limited budget. Active learning has been successfully used in several tasks including recommender systems (Rubens et al., 2015), sound event detection (Shuyang et al., 2020), analysis of driving time series (Jarl et al., 2022), reaction prediction in drug discovery (Viet Johansson et al., 2022) and logged data analysis (Yan et al., 2018). Within active learning, the querying strategy is performed using an *acquisition function*.

Active learning has been applied to clustering as well, and is sometimes called *supervised clustering* (Awasthi & Zadeh, 2010), where the goal is to discover the ground-truth clustering with a minimal number of queries. In this setting, the queries are generally performed in one of two ways: i) Queries asking if two clusters should *merge* or if one cluster should be *split* into multiple clusters (Balcan & Blum, 2008; Awasthi & Zadeh, 2010; Awasthi et al., 2017). ii) Queries for the pairwise relations between objects (Basu et al., 2004; Eriksson et al., 2011; Krishnamurthy et al., 2012; Bonchi et al., 2013a; Korlakai Vinayak & Hassibi, 2016; Ailon et al., 2018; Mazumdar & Saha, 2017b;a; Saha & Subramanian, 2019; Bressan et al., 2019; García-Soriano et al., 2020; van Craenendonck et al., 2018b; Soenen et al., 2021).

Among the aforementioned active learning works, only (Mazumdar & Saha, 2017b; Bressan et al., 2019; García-Soriano et al., 2020) are in the setting that we consider: i) the clustering algorithm used is based on correlation clustering, ii) the pairwise similarities are not assumed to be known in advance and iii) we do not assume access to feature vectors (i.e., we only get information about the ground-truth clustering by querying the oracle for pairwise relations).[1] All of these studies propose algorithms that are derived from a theoretical perspective, resulting in multiple practical restrictions. The work in (García-Soriano et al., 2020) introduces a *pivot-based* query-efficient correlation clustering algorithm called QECC. It starts by randomly selecting a pivot object $u$ and then queries the similarity between $u$ and all other (unclustered) objects to form a cluster with $u$ and the objects with a positive similarity with $u$. It then continues iteratively with the remaining objects until all objects are assigned to a cluster or the querying budget is reached. The method in (Bressan et al., 2019) is very similar to QECC, except that after selecting a pivot $u$ it only queries the similarity between $u$ and a random subset $S$ of all other objects. Only if there exists an object in $S$ with a positive similarity to $u$ will it query the remaining objects in order to form a cluster. Otherwise, the pivot $u$ is assumed to be a singleton cluster. Finally, (Mazumdar & Saha, 2017b) develop a number of more complex pivot-based algorithms that satisfy theoretical guarantees on the query complexity, assuming a noisy oracle. However, the algorithms are purely theoretical and are not implemented and investigated in practice, and require setting a number of non-trivial parameters (e.g., they assume the noise level is known in advance).

These works on active correlation clustering suffer from various limitations: i) They only consider binary pairwise similarities in $\{-1, 1\}$, whereas, as extensively discussed in the correlation clustering literature (e.g., (Demaine et al., 2006; Bun et al., 2021; Ailon et al., 2008)), having real-number pairwise similarities provides more flexibility, informativeness and tolerance to noise. ii) They do not follow the generic active learning procedure as they assume the querying process is tightly integrated into the clustering algorithm itself. In fact, the pairwise relations to be queried are selected randomly without considering the notion of informativeness. iii) They are severely affected by the presence of noise in the oracle. While the algorithms in (Mazumdar & Saha, 2017b) are meant to be robust to noise, they are quite limited in practice. For example, they rely on precisely defined constants and the true noise level is assumed to be known in advance. iv) Only pivot-based active correlation clustering methods (i.e., (Bressan et al., 2019; García-Soriano et al., 2020)) have been investigated. However, as studied for non-active (standard) correlation clustering (Thiel et al., 2019), pivot-based methods yield poor clustering results. This is consistent with our experimental studies in the active correlation clustering setting (Section 4).

---

[1]The works in (Basu et al., 2004; van Craenendonck et al., 2018b; Soenen et al., 2021) are developed for *active constraint clustering*. We adapt the latter two to our setting and investigate them in our experiments.

Motivated by these limitations, we develop a generic framework for active learning for correlation clustering that better resembles the standard active learning framework. First, the pairwise similarities can be any positive or negative real number, and might even be inconsistent (i.e., violate transitivity). This yields full flexibility in the type of feedback that a user or annotator can provide, in particular in uncertain settings. For example, assume the true similarity between objects $u$ and $v$ is $+1$. With binary feedback, the annotator may simply return $-1$ in the case of uncertainty/mistake. However, with feedback in $[-1, 1]$ their mistake/uncertainty may yield for example $-0.1$. A faulty feedback of $-0.1$ is much less severe than $-1$, and our framework can take advantage of this. Second, the process of querying pairwise similarities is separated from the clustering algorithm. This leads to more flexibility in how the query strategies can be constructed and they can be used in conjunction with *any* correlation clustering algorithm. In particular, we adapt an efficient correlation clustering method based on local search whose effectiveness has been demonstrated in standard correlation clustering setting (Thiel et al., 2019; Chehreghani, 2022). Third, the queries are robust w.r.t. a noisy oracle by allowing multiple queries for the same pairwise similarity (i.e., a non-persistent noise model is considered). Finally, it automatically identifies the number of clusters based on the available pairwise similarities. To the best of our knowledge, this framework is the first work that fulfills all the above-mentioned advantages.

In addition to the generic active clustering framework, we propose and analyze a number of novel acquisition functions (query strategies) suited for correlation clustering. In particular, we analyze the triangles of objects and their relations in correlation clustering and propose two novel query strategies called *maxmin* and *maxexp*. We finally demonstrate the effectiveness of the framework and the proposed query strategies via several experimental studies.

## 2 Active Correlation Clustering

In this section, we introduce our framework for active learning for correlation clustering. We start with the problem formulation, then describe the respective active learning procedure, and finally explain the clustering algorithm to be used in our framework.

### 2.1 Problem formulation

We are given a set of $N$ objects (data points) denoted by $\boldsymbol{V} = \{1, \ldots, N\}$. The set of the pairs of objects in $\boldsymbol{V}$ is denoted by $\boldsymbol{E} = \{(u, v) \mid u, v \in \mathbf{V} \text{ and } u > v\}$ and will be used to indicate the pairwise relations between objects. Thus, we have $|\boldsymbol{E}| = \binom{N}{2}$.

We assume there exists a generic similarity function $\sigma^* : \boldsymbol{E} \to \mathbb{R}$ representing the true pairwise similarities between every pair $(u, v) \in \boldsymbol{E}$. However, this similarity function is not known in advance; one can only query the oracle for a noisy instantiation of this function for a desired pair of objects while incurring some cost. We then use $\sigma : \boldsymbol{E} \to \mathbb{R}$ to refer to some estimate of the pairwise similarities. If $\sigma(u, v) = \sigma^*(u, v)$ for all $(u, v) \in \boldsymbol{E}$ we have a perfect estimate of the true pairwise similarities, which is unrealistic in practice. Therefore, the goal is to recover the ground-truth clustering with a minimal number of (active) queries for the pairwise similarities to the oracle, as each query incurs some cost.[2]

The set $\boldsymbol{V}$ together with a similarity function $\sigma$ can be viewed as a weighted (undirected) graph $\mathcal{G}_\sigma = (\boldsymbol{V}, \boldsymbol{E})$ where the weight of edge $(u, v) \in \boldsymbol{E}$ is $\sigma(u, v)$. We denote by $e = (u, v)$ an edge in $\boldsymbol{E}$. Let $\mathcal{G}_\sigma(\boldsymbol{z}) = (\boldsymbol{z}, \boldsymbol{E_z})$ denote the subgraph of $\mathcal{G}_\sigma$ induced by $\boldsymbol{z} \subseteq \boldsymbol{V}$, where $\boldsymbol{E_z} = \{(u, v) \in \boldsymbol{E} \mid u, v \in \boldsymbol{z} \text{ and } u > v\} \subseteq \boldsymbol{E}$. A clustering of $\boldsymbol{z} \subseteq \boldsymbol{V}$ is a partition of $\boldsymbol{z}$ into $k$ disjoint sets (clusters). Let $\mathcal{C_z}$ be the set of all clusterings of $\boldsymbol{z}$. Consider a clustering $C \in \mathcal{C_z}$. According to correlation clustering, an edge $(u, v) \in \boldsymbol{E_z}$ violates the clustering $C$ if $\sigma(u, v) \geq 0$ while $u$ and $v$ are in different clusters or if $\sigma(u, v) < 0$ while $u$ and $v$ are in the

---

[2]Querying an edge in $\mathbf{E}$ means querying its weight, i.e., the pairwise similarity that it represents.

same cluster.[3] Thus, given $C$ and $\sigma$, the function of cluster violations $\boldsymbol{R}_{(\boldsymbol{z},\sigma,C)} : \boldsymbol{E_z} \to \mathbb{R}^+$ is defined as

$$\boldsymbol{R}_{(\boldsymbol{z},\sigma,C)}(u,v) = \begin{cases} |\sigma(u,v)|, & \text{iff } (u,v) \text{ violates } C \\ 0, & \text{otherwise} \end{cases} \tag{1}$$

Then, we define the correlation clustering cost function $\Delta_{(\boldsymbol{z},\sigma)} : \mathcal{C}_{\boldsymbol{z}} \to \mathbb{R}^+$ given a similarity function $\sigma$ as

$$\Delta_{(\boldsymbol{z},\sigma)}(C) = \sum_{(u,v) \in \mathbf{E_z}} \mathbf{R}_{(\mathbf{z},\sigma,C)}(u,v). \tag{2}$$

We aim for a clustering that is maximally consistent with a set of (provided) feedbacks $\sigma$. The optimal clustering of $\boldsymbol{z}$ given $\sigma$ is $C^*_{(\boldsymbol{z},\sigma)} = \arg\min_{C \in \mathcal{C}_{\boldsymbol{z}}} \Delta_{(\boldsymbol{z},\sigma)}(C)$. Thus, given $\sigma^*$ the optimal correlation clustering solution w.r.t. the true pairwise similarities would be $C^*_{(\boldsymbol{V},\sigma^*)}$. In this work, we consider the case when $\sigma^*$ is unknown and aim to estimate it appropriately via active learning in order to perform a proper correlation clustering. Finally, we let $\mathcal{A}(\boldsymbol{z},\sigma) \in \mathcal{C}_{\boldsymbol{z}}$ be the clustering found by some correlation clustering algorithm $\mathcal{A}$ given $\mathcal{G}_\sigma(\boldsymbol{z})$.

## 2.2 Active correlation clustering procedure

---

**Algorithm 1** Active clustering procedure

---

    **Input**: Data objects $\boldsymbol{V}$, initial similarity function $\sigma_0$, clustering algorithm $\mathcal{A}$, query strategy $\mathcal{S}$, noise level $\gamma \in [0,1]$, batch size $B$
    **Output**: Clustering $C \in \mathcal{C}_{\boldsymbol{V}}$
1: $\mathcal{Q}^0 \leftarrow \{\emptyset\}^{|\boldsymbol{V}| \times |\boldsymbol{V}|}$
2: $\mathcal{Q}^0_{uv} \leftarrow \mathcal{Q}^0_{uv} \cup \sigma_0(u,v), \ \forall (u,v) \in \boldsymbol{E}$
3: $i \leftarrow 0$
4: **while** stopping criterion is not met **do**
5:     $C^i \leftarrow \mathcal{A}(\boldsymbol{V}, \sigma_i)$
6:     $\mathcal{B} \leftarrow \mathcal{S}(C^i, \mathcal{Q}^i, \sigma_i, B)$                                                       $\triangleright |\mathcal{B}| = \mathrm{B}$
7:     **for** $(u,v) \in \mathcal{B}$ **do**
8:         $Q \leftarrow \text{ORACLE}(u,v,\gamma)$
9:         $\mathcal{Q}^{i+1}_{uv} \leftarrow \mathcal{Q}^i_{uv} \cup Q$
10:        $\sigma_{i+1}(u,v) \leftarrow \frac{1}{|\mathcal{Q}^{i+1}_{uv}|} \sum_{q \in \mathcal{Q}^{i+1}_{uv}} q$
11:     **end for**
12:     $i \leftarrow i+1$
13: **end while**
14: Return $C^i$

---

Algorithm 1 outlines the active correlation clustering procedure. It takes as input the set of objects $\boldsymbol{V}$, an initial similarity function $\sigma_0$, a correlation clustering algorithm $\mathcal{A}$ and a query strategy $\mathcal{S}$. The initial similarity function can contain partial or no information about $\sigma^*$, depending on the initialization method. We will assume that $|\sigma_0(e)| \leq \lambda$ for all $e \in \mathbf{E}$, where $\lambda \in [0,1]$. In this paper, $\lambda = 0.1$ as this indicates that we are not confident about the initial edge weights. The algorithm begins by initializing a query matrix $\mathcal{Q}^i$ where $\mathcal{Q}^i_{uv}$ is a set containing all queries made for edge $(u,v) \in \boldsymbol{E}$ until iteration $i$. Each iteration $i$ consists of three steps. First, it runs the clustering algorithm $\mathcal{A}$ on objects $\boldsymbol{V}$ given the current similarity function $\sigma_i$. This returns the clustering $C^i \in \mathcal{C}_{\boldsymbol{V}}$. Second, the query strategy $\mathcal{S}$ selects a batch $\mathcal{B} \subseteq \boldsymbol{E}$ of $B$ edges. Then, the oracle $\mathcal{O}$ is queried for the weights of all the edges $(u,v) \in \mathcal{B}$. In this paper, we consider a noisy oracle where the noise is non-persistent. This means that each query for $(u,v) \in \boldsymbol{E}$ returns $\sigma^*(u,v)$ with probability $1-\gamma$ or a value from the set $[-1,-\lambda) \cup (\lambda,1]$ uniformly at random with probability $\gamma$. This noise

---

[3]$\sigma(u,v) = 0$ implies neutral feedback, where the oracle cannot decide if the objects are similar or dissimilar. For brevity, we treat this case similarly to positive pairwise similarities.

model ensures that querying the same edge multiple times may be beneficial (which models an oracle that is allowed to correct for previous mistakes) and that noisy queries will not be smaller in absolute value than the initial edge weights in $\sigma_0$. This will lead to better exploration for some of the query strategies proposed in Section 3. In addition, we assume that the cost of each query (to different pairs) is equal (e.g., 1), though our framework is generic enough to encompass a varying query cost as well (where the acquisition function is divided by the respective cost). This choice means that the problem reduces to minimizing the number of queries made. This is the setting studied in most of the prior active learning work. Finally, the current similarity function $\sigma_i$ is updated based on the queries to the oracle by setting $\sigma_{i+1}(u, v)$ equal to the average of all queries made for the edge $(u, v)$ so far. This leads to robustness against noise since a noisy query of an edge $(u, v)$ will have less impact in the long run.

### 2.3 Active correlation clustering with zero noise

If a given similarity function $\sigma$ is fully consistent (which means that there is no noise flipping the sign of pairwise similarities and the transitive property of the pairwise similarities is satisfied), there will always exist a solution for the correlation clustering objective in Eq. 2 with the cost of zero, and finding the solution is no longer NP-hard (unlike general correlation clustering). It is sufficient to simply extract the edges $(u, v) \in \mathbf{E}$ such that $\sigma(u, v) > 0$ and assign objects to their corresponding clusters. Therefore, in this paper, we consider the more challenging *noisy* setting where the information obtained from the oracle is noisy and leads to inconsistency among pairwise relations.

Most of the previous studies on active clustering that query pairwise relations between objects consider the zero noise setting and utilize methods that fully exploit this assumption. This is generally done by using the queried pairwise similarities to infer the pairwise relations that have not been queried yet, which may lead to improvement in performance in the zero noise setting. Let $\mathbf{E}_{\text{queried}} \subseteq \mathbf{E}$ be the set of edges that have been queried so far. Consider the three objects $u, v, w \in \mathbf{V}$, where $(u, v), (u, w) \in \mathbf{E}_{\text{queried}}$, $(v, w) \in \mathbf{E} \setminus \mathbf{E}_{\text{queried}}$, $\sigma_i(u, v) = 1$, $\sigma_i(u, w) = 1$. Given this, one can easily infer $\sigma_i(v, w) = 1$ using the transitive property of the pairwise similarities. In principle, one can perform a multi-level inference where previously inferred information is used to infer new information. These methods are highly sensitive to noise and will in general perform very badly with even a small amount of noise since propagating noisy information in this manner cannot be recovered from. Therefore, they are not suitable for a *noisy* setting.

### 2.4 Correlation clustering algorithm

Unlike the previous active correlation clustering methods (Mazumdar & Saha, 2017b; Bressan et al., 2019; García-Soriano et al., 2020) that are bound to specific correlation clustering algorithms, our framework is generic in the sense that it can be employed with *any* arbitrary correlation clustering method. Thereby, we can use any of the several approximate algorithms proposed in the literature, e.g., (Bansal et al., 2004; Demaine et al., 2006; Ailon et al., 2008; Charikar et al., 2005; Elsner & Schudy, 2009; Giotis & Guruswami, 2006). Among them, the methods based on *local search* perform significantly better in terms of both the quality of clustering and the computational runtimes (Thiel et al., 2019; Chehreghani, 2022). Thereby, we employ a local search approach to solve the correlation clustering problem for a pairwise similarity matrix $\sigma$.[4] The local search method in (Chehreghani, 2022) assumes a fixed number of clusters for correlation clustering. However, given the pairwise similarities $\sigma$, the minimal violation objective should automatically determine the most consistent number of clusters. Consider, for example, the case where all the pairwise similarities are negative. Then, the clustering algorithm should place every object in a separate cluster. On the other hand, when all the pairwise similarities are positive, then only one cluster must be identified from the data.

The work in (Thiel et al., 2019) develops a Frank-Wolfe optimization approach to correlation clustering, which then leads to a local search method with an optimized choice of the update parameter. To deal with the non-fixed (data-dependent) number of clusters, it sets the initial number of clusters to be equal to the number of objects $N$, and lets the algorithm try all numbers and thus leave some clusters empty. This

---

[4]The effectiveness of greedy methods based on local search has been studied extensively in several other clustering settings as well, e.g., (Dhillon et al., 2004; 2005).

method yields some unnecessary extra computation that leads to increasing the overall runtime. Thereby, in this paper, we develop a local search method for correlation clustering that at each local search step, automatically increases or decreases the number of clusters only if needed. In this method, we begin with a randomly initialized clustering solution and then we iteratively assign each object to the cluster that achieves a maximal reduction in the violations (i.e., the cost function in Eq. 2 is maximally reduced). We allow for the possibility that the object moves to a new (empty) cluster, and in this way, we provide the possibility to expand the number of clusters. On the other hand, when the last object in an existing cluster leaves to move to another cluster, then the cluster disappears. We repeat this local search procedure until no further changes (in the assignments of objects to clusters) happen for an entire round of investigating the objects. In this case, a local optimal solution is attained and we stop the local search. We may repeat the local search procedure multiple times and choose the solution with a minimal cost.

A naive implementation of this method would require a time complexity of $\mathcal{O}(kN^3)$ (assume $k$ is the number of clusters). We instead use an efficient algorithm whose time complexity is $\mathcal{O}(kN^2)$. Given clustering $C$, assume $C_u$ indicates the cluster label for object $u$. Then, similar to the shifted Min Cut or correlation clustering with a fixed number of clusters in (Chehreghani, 2013; 2022), we can write the cost function in Eq. 2 as

$$
\begin{aligned}
\Delta_{(\mathbf{z},\sigma)}(C) &= \sum_{(u,v)\in\mathbf{E_z}} \mathbf{R}_{(\mathbf{z},\sigma,C)}(u,v) \\
&= \sum_{\substack{(u,v)\in\mathbf{E_z} \\ C_u=C_v}} \frac{1}{2}(|\sigma(u,v)| - \sigma(u,v)) + \sum_{\substack{(u,v)\in\mathbf{E_z} \\ C_u\neq C_v}} \frac{1}{2}(|\sigma(u,v)| + \sigma(u,v)) \\
&= \frac{1}{2}\sum_{(u,v)\in\mathbf{E_z}} |\sigma(u,v)| - \frac{1}{2}\sum_{\substack{(u,v)\in\mathbf{E_z} \\ C_u=C_v}} \sigma(u,v) + \frac{1}{2}\sum_{(u,v)\in\mathbf{E_z}} \sigma(u,v) - \frac{1}{2}\sum_{\substack{(u,v)\in\mathbf{E_z} \\ C_u=C_v}} \sigma(u,v) \\
&= \underbrace{\frac{1}{2}\sum_{(u,v)\in\mathbf{E_z}} (|\sigma(u,v)| + \sigma(u,v))}_{\text{constant}} - \sum_{\substack{(u,v)\in\mathbf{E_z} \\ C_u=C_v}} \sigma(u,v).
\end{aligned}
\tag{3}
$$

The first term in Eq. 3 is *constant* w.r.t. the choice of a particular clustering $C$. Thereby, we have

$$
\Delta_{(\mathbf{z},\sigma)}(C) = -\Delta_{(\mathbf{z},\sigma)}^{\text{MaxCor}}(C) + \text{constant},
\tag{4}
$$

where the *max correlation* objective function $\Delta_{(\mathbf{z},\sigma)}^{\text{MaxCor}} : \mathcal{C_z} \rightarrow \mathbb{R}$ is defined as

$$
\Delta_{(\mathbf{z},\sigma)}^{\text{MaxCor}}(C) = \sum_{\substack{(u,v)\in\mathbf{E_z} \\ C_u=C_v}} \sigma(u,v).
\tag{5}
$$

Then, we have

$$
\underset{C\in\mathcal{C_z}}{\arg\max}\ \Delta_{(\mathbf{z},\sigma)}^{\text{MaxCor}}(C) = \underset{C\in\mathcal{C_z}}{\arg\min}\ \Delta_{(\mathbf{z},\sigma)}(C).
\tag{6}
$$

Therefore, we will seek to maximize $\Delta_{(\mathbf{z},\sigma)}^{\text{MaxCor}}(C)$, while the number of clusters is non-fixed and dynamic.

Assume in clustering $C$ the cluster label of object $u$ is $l$, i.e., $C_u = l$. Then, $\Delta_{(\mathbf{z},\sigma)}^{\text{MaxCor}}(C)$ is written as

$$\Delta^{\mathrm{MaxCor}}_{(\mathbf{z},\sigma)}(C) = \sum_{\substack{(v,w)\in\mathbf{E}_{\mathbf{z}} \\ C_v=C_w}} \sigma(v,w)$$

$$= \underbrace{\sum_{\substack{(v,w)\in\mathbf{E}_{\mathbf{z}} \\ C_v=C_w \\ v,w\neq u}} \sigma(v,w)}_{\mu} + \underbrace{\sum_{\substack{v\in\mathbf{z} \\ C_v=l}} \sigma(u,v)}_{\delta}, \tag{7}$$

where the first term (i.e., $\mu$) is independent of $u$ and only the second term $\delta$ depends on $u$.

As mentioned, we perform a local search in order to minimize the cost function $\Delta_{(\mathbf{z},\sigma)}(C)$ or equivalently maximize the objective $\Delta^{\mathrm{MaxCor}}_{(\mathbf{z},\sigma)}(C)$. In this method, we start with a random clustering solution and then we iteratively assign each object to the cluster that yields a maximal reduction in the cost function or a maximal improvement of the objective $\Delta^{\mathrm{MaxCor}}_{(\mathbf{z},\sigma)}(C)$. Consider an intermediate clustering $C$ with $k$ clusters in an arbitrary step of local search. To compute the new cluster of an object $u$, one needs to evaluate the cost/objective function for different clusters and choose the cluster which leads to maximal improvement. Assuming the clustering is performed on all the objects in $\mathbf{V}$. One evaluation of the cost function $\Delta_{(\mathbf{V},\sigma)}(C)$ or the objective function $\Delta^{\mathrm{MaxCor}}_{(\mathbf{V},\sigma)}(C)$ would have the computational complexity $\mathcal{O}(N^2)$. Since this evaluation is performed for each of $k$ clusters (i.e., $u$ is investigated to be assigned to each of $k$ clusters), then the naive implementation of the method would have the complexity of $\mathcal{O}(kN^2)$. If this is repeated for all objects, then it will be $\mathcal{O}(kN^3)$. However, introducing $\Delta^{\mathrm{MaxCor}}_{(\mathbf{z},\sigma)}(C)$ and in particular its formulation in Eq. 7 provides a computationally efficient way to implement the local search, similar to the Shifted Min Cut method in (Chehreghani, 2022). For each object $u$, only the second term $\delta$ in Eq. 7 varies as we investigate different clusters for this object. Thus, the first term (i.e., $\mu$) is the same when evaluating the assignment of object $u$ to different clusters and can be discarded. In this way, evaluating an object for $k$ different clusters will have the computational complexity of $\mathcal{O}(kN)$ (instead of $\mathcal{O}(kN^2)$), and hence the complexity for all the objects becomes $\mathcal{O}(kN^2)$ (instead of $\mathcal{O}(kN^3)$).

This local search method is outlined in Algorithm 2. It takes as input a set of objects $\mathbf{z}$, a similarity function $\sigma$, an initial number of clusters $k$, the number of repetitions $T$, and a stopping threshold $\eta$. In our experiments, we set $T = 3$, $\eta = 2^{-52}$ (double precision machine epsilon) and $k = |\mathbf{z}|$ in the first iteration of Algorithm 1, and then $k = |C^i|$ for all remaining iterations where $|C^i|$ denotes the number of clusters in the current clustering $C^i$ (in Algorithm 1). The output is a clustering $C \in \mathcal{C}_{\mathbf{z}}$. The main part of the algorithm (lines 6-27) is based on the local search of the respective non-convex objective. Therefore, we run the algorithm $T$ times with different random initializations and return the best clustering in terms of the objective function. The main algorithm (starting from line 6) consists of initializing the current clustering $C$ randomly. Then, it loops for as long as the current *max correlation* objective changes by at least $\eta$ compared to the last iteration. If not, we assume it has converged to some (local) optimum. Each repetition consists of iterating over all the objects in $\mathbf{z}$ in a random order $\mathbf{O}$ (this ensures variability between the $T$ runs). For each object $u \in \mathbf{O}$, it calculates the similarity (correlation) between $u$ and all clusters $c \in C$, which is denoted by $\mathbf{S}_c(u)$. $\mathbf{S}_c(u)$ is computed based on the second term (i.e., $\delta$) in Eq. 7. Then, the cluster that is most similar to $u$ is obtained by $\arg\max_{c \in C} \mathbf{S}_c(u)$. Now, if the most similar cluster to $u$ has a negative correlation score, this indicates that $u$ is not sufficiently similar to any of the existing clusters. Thus, we construct a new cluster with $u$ as the only member. If the most similar cluster to $u$ is positive, we simply assign $u$ to this cluster. Thus, the number of clusters will dynamically change based on the pairwise similarities (it is possible that the only object of a singleton cluster is assigned to another cluster and thus the singleton cluster disappears). Finally, in each repetition the current *max correlation* objective is computed efficiently by only updating it based on the current change of the clustering $C$ (i.e., lines 19 and 25).

## 3 Query Strategies

In this section, we describe and investigate a number of query strategies for active learning applied to correlation clustering. A simple baseline query strategy would be to query the weight of the edge $e \in \mathbf{E}$

---

**Algorithm 2** Max Correlation Clustering Algorithm $\mathcal{A}$ (dynamic $k$)

---

**Input**: Objects $\mathbf{z} \subseteq \mathbf{V}$, similarity function $\sigma$, initial number of clusters $k$, number of iterations $T$, stopping threshold $\eta$

**Output**: Clustering $C \in \mathcal{C}_{\mathbf{z}}$

1: $N \leftarrow |\mathbf{z}|$
2: $k \leftarrow \min(k, N)$
3: $\Delta_{\text{best}} \leftarrow -\infty$
4: $C_{\text{best}} \leftarrow \{\mathbf{z}\}$
5: **for** $j \in \{1, \ldots, T\}$ **do**
6:     $C \leftarrow \{c^1, \ldots, c^k\}$, where $c^i = \{\emptyset\}$ for all $i \in \{1, \ldots, k\}$
7:     Assign each object in $\mathbf{z}$ to one of the clusters in $C$ uniformly at random
8:     $\Delta \leftarrow \Delta_{(\mathbf{z}, \sigma)}^{\text{MaxCor}}(C)$
9:     $\Delta_{\text{old}} \leftarrow \Delta - 1$
10:     **while** $|\Delta - \Delta_{old}| > \eta$ **do**
11:         $\Delta_{old} \leftarrow \Delta$
12:         $\mathbf{O} \leftarrow$ a random permutation of the objects in $\mathbf{z}$
13:         **for** each $u$ in $\mathbf{O}$ **do**
14:             $\mathbf{S}_c(u) \leftarrow \sum_{v \in c, u \neq v} \sigma(u, v)$ for all $c \in C$
15:             $c_{\max} \leftarrow \arg\max_{c \in C} \mathbf{S}_c(u)$
16:             $c_u \leftarrow$ cluster $c \in C$ that currently involves $u$
17:             **if** $\mathbf{S}_{c_{\max}}(u) < 0$ **then**
18:                 Remove $u$ from its current cluster $c_u$
19:                 $\Delta \leftarrow \Delta - \mathbf{S}_{c_u}(u)$
20:                 $c_{\text{new}} \leftarrow \{u\}$
21:                 $C \leftarrow C \cup c_{\text{new}}$
22:                 $k \leftarrow k + 1$
23:             **else**
24:                 Move $u$ from its current cluster to $c_{\max}$
25:                 $\Delta \leftarrow \Delta - \mathbf{S}_{c_u}(u) + \mathbf{S}_{c_{\max}}(u)$
26:                 Remove empty clusters from $C$ and decrease $k$ accordingly
27:             **end if**
28:         **end for**
29:     **end while**
30:     **if** $\Delta > \Delta_{\text{best}}$ **then**
31:         $C_{\text{best}} \leftarrow C$
32:         $\Delta_{\text{best}} \leftarrow \Delta$
33:     **end if**
34: **end for**
35: **return** $C_{\text{best}}$

---

selected uniformly at random. However, this strategy may not lead to querying the most informative edge weights. Instead, one can construct more sophisticated query strategies based on information in one or more of the following three components: the current clustering $C^i$, the current query matrix $\mathcal{Q}^i$ and the current similarity function $\sigma_i$.

## 3.1 Uncertainty and frequency

In this section, we propose two simple query strategies that are meant to be used as baselines. Uncertainty usually refers to lack of knowledge. Therefore, one way to quantify uncertainty is to consider the edge with the smallest edge weight in absolute value, since a smaller magnitude indicates lack of confidence. In other words, we select and query the edge according to

$$\hat{e} = \underset{e \in \boldsymbol{E}}{\arg \min} |\sigma_i(e)|. \tag{8}$$

However, this may not always be an optimal strategy since a low magnitude of an edge weight does not necessarily imply informativeness. To see this, consider three distinct nodes $(u, v, w) \subseteq \boldsymbol{V}$ with pairwise similarities $\{\sigma(u,v), \sigma(u,w), \sigma(v,w)\} = \{1, 1, 0.1\}$. While $\sigma(v,w)$ has a small absolute value, one can infer that its value must be 1 using the transitive property of pairwise similarities (see Section 3.2). Thus, querying the edge $(v,w)$ is unlikely to impact the resulting clustering.

An alternative strategy, called *frequency*, aims to query the edge with the smallest number of queries so far, i.e.,

$$\hat{e} = \underset{e \in \boldsymbol{E}}{\arg \min} |\mathcal{Q}_e^i|. \tag{9}$$

Eq. 8 and Eq. 9 will behave very similarly if the pairwise similarities in the initial similarity matrix $\sigma_0$ are small in absolute value, e.g., $|\sigma_0(e)| \leq 0.1$ for all $e \in \boldsymbol{E}$. Additionally, with a more noisy oracle, Eq. 8 may query the same edge more than once (which could be beneficial), although this is quite unlikely. Finally, a batch of edges $\mathcal{B} \subseteq \boldsymbol{E}$ can be selected by selecting the top $|\mathcal{B}|$ edges from Eq. 8 or Eq. 9.

### 3.2 Inconsistency and the maxmin query strategy

Inconsistency indicates that there is no clustering $C \in \mathcal{C}_{\boldsymbol{z}}$ of some $\boldsymbol{z} \subseteq V$ such that the correlation clustering cost $\Delta_{(\boldsymbol{z}, \sigma)}(C)$ is 0. This occurs when the similarity function $\sigma$ consists of conflicting information about how a particular edge $(u,v) \in \boldsymbol{E}_{\boldsymbol{z}}$ is to be clustered. In turn, this happens when $\sigma$ does not satisfy the transitive property of the pairwise similarities. Sets of *three* objects $(u, v, w) \subseteq \boldsymbol{V}$, which we call *triangles*, are the smallest collection of objects for which an inconsistency can occur. The transitive property says that if $\sigma(u,v) \geq 0$ and $\sigma(u,w) \geq 0$ then $\sigma(v,w) \geq 0$ or if $\sigma(u,v) \geq 0$ and $\sigma(u,w) < 0$ then $\sigma(v,w) < 0$. The inconsistencies in a set of four or more objects in $\boldsymbol{V}$ come down to considering each of the triangles that can be constructed from the objects in the set. We discuss this in more detail later, but it highlights the importance of triangles. We denote by $\boldsymbol{T}$ the set of triangles $t = (u, v, w)$ of distinct objects in $\boldsymbol{V}$, i.e., $|\boldsymbol{T}| = \binom{|\boldsymbol{V}|}{3}$, and $\boldsymbol{E}_t = \{(u,v), (u,w), (v,w)\}$ indicates the edges between the objects of triangle $t = (u, v, w) \in \boldsymbol{T}$.

In this section, we propose a query strategy that queries the weight of an edge in some triangle $t \in \boldsymbol{T}$ with the smallest contribution to the inconsistency of $t$. Such an edge represents the *minimal* correction of the edge weights in the triangle $t$ in order to satisfy consistency. Furthermore, since we are in an active learning setting, we want to choose the *maximum* of such corrections among all the triangles (to gain maximal information/correction). We call this query strategy *maxmin*. Theorem 1 provides a way to select an edge according to the maxmin query strategy.

**Theorem 1.** *Given $\sigma$, let $\mathcal{T}_\sigma \subseteq \boldsymbol{T}$ be the set of triangles $t = (u, v, w) \in \boldsymbol{T}$ with exactly two positive edge weights and one negative edge weight. Then, the maxmin query strategy corresponds to querying the weight of the edge $\hat{e}$ selected by*

$$(\hat{t}, \hat{e}) = \underset{t \in \mathcal{T}_\sigma}{\arg \max} \underset{e \in \boldsymbol{E}_t}{\min} |\sigma(e)|. \tag{10}$$

*Proof.* See Appendix A.1. □

The set $\mathcal{T}_\sigma$ introduced in Theorem 1 will be referred to as the set of *bad triangles* (given $\sigma$). The reason is that for every $t \in \mathcal{T}_\sigma$ there is no clustering $C \in \mathcal{C}_t$ such that $\Delta_{(t,\sigma)}(C) = 0$ (i.e., they induce inconsistency). On the other hand, $\boldsymbol{T} \setminus \mathcal{T}_\sigma$ will be referred to as the set of *good triangles*, i.e., those for which there is a clustering $C \in \mathcal{C}_t$ such that $\Delta_{(t,\sigma)}(C) = 0$.

We now provide some further motivation for why we consider triangles (i.e., collections of three objects) instead of collections of four or more objects in $\boldsymbol{V}$. We say that two triangles $t_1, t_2 \in \boldsymbol{T}$ are correlated if they share an edge. A lower bound of $\min_{C \in \mathcal{C}_{\boldsymbol{V}}} \Delta_{(\boldsymbol{V}, \sigma)}(C)$ can be obtained by summing the minimal cost

$\min_{C \in \mathcal{C}_t} \Delta_{(t,\sigma)}(C)$ of all edge-disjoint (uncorrelated) bad triangles. This is because all uncorrelated bad triangles $t$ must contain at least one unique edge $(u,v) \in \boldsymbol{E}_t$ that violates the clustering. It is a lower bound since additional mistakes can occur in any triangle $t \in \boldsymbol{T}$ (whether it is good or bad) if it is correlated with one or more bad triangles. Thus, a correlation clustering algorithm $\mathcal{A}$ must consider such correlations in order to find a clustering that best balances all of the (possibly conflicting) inconsistencies. The inconsistencies that can occur among collections of four or more objects in $\boldsymbol{V}$ boil down to considering the (potentially correlated) triangles that can be constructed from the objects in the collection.

Thereby, it is clear that maxmin prefers to select edges in bad triangles. Assuming $\mathcal{T}_{\sigma^*} = \emptyset$, if $\mathcal{T}_{\sigma_i} \neq \emptyset$ in some iteration $i$ (due to a poor initialization of $\sigma_0$ or a noisy oracle), then all bad triangles $t \in \mathcal{T}_{\sigma_i}$ are guaranteed to contain at least one edge $e \in \boldsymbol{E}_t$ with a similarity $\sigma_i(e)$ that does not agree with $\sigma^*(e)$ (i.e., wrong information). Therefore, it would be beneficial to query the edges in bad triangles in order to reveal $\sigma^*$ as quickly as possible.

We observe that maxmin would prefer a bad triangle with the edge weights $\{1, 0.7, -0.7\}$ over a bad triangle with the edge weights $\{1, 1, -0.1\}$. The reason is that the first triangle is more confidently inconsistent compared to the second one. This means that the clustering of the three objects involved in the first triangle will increase the value of the clustering cost $\Delta_{(\boldsymbol{V},\sigma)}$ more than the second triangle (also see proof of Theorem 1 for more on this). Therefore, it makes sense to prioritize querying one of the edges in the first triangle to resolve the inconsistency.

Finally, since maxmin ranks all good triangles equally we will follow the random query strategy to select an edge if $\mathcal{T}_{\sigma_i} = \emptyset$ in some iteration $i$. Also, a batch of edges $\mathcal{B} \subseteq \boldsymbol{E}$ can be selected by selecting the top $|\mathcal{B}|$ edges from Eq. 10 (see Section 3.5 for details).

### 3.3 Maxexp query strategy

Consider a triangle $(u, v, w)$ with edge weights $\{\sigma(u,v), \sigma(u,w), \sigma(v,w)\}$, where $\sigma(u,v)$ is fixed, $|\sigma(u,w)| \geq |\sigma(u,v)|$ and $|\sigma(v,w)| \geq |\sigma(u,v)|$. Then, the maxmin strategy will give equal scores to all such triangles regardless of what $\sigma(u,w)$ and $\sigma(v,w)$ are, since $|\sigma(u,v)|$ is always the smallest edge weight in absolute value. Hence, a triangle $t_1$ with edge weights $\{1, 1, -0.1\}$ will be ranked equally to a triangle $t_2$ with edge weights $\{0.1, 0.1, -0.1\}$. We would like for $t_1$ to be selected over $t_2$ since it is more confidently inconsistent. This issue occurs because maxmin discards all edges except for the edge with minimal weight in absolute value. One way to fix this is to replace the smallest absolute edge weight (i.e., the *minimal* inconsistency) with the *expected* inconsistency or violation. We call this query strategy *maxexp*.

Given a triangle $t \in \boldsymbol{T}$, maxexp defines the probability of a clustering $C \in \mathcal{C}_t$ as

$$p(C|t) = \frac{\exp(-\beta \Delta_{(t,\sigma)}(C))}{\sum_{C' \in \mathcal{C}_t} \exp(-\beta \Delta_{(t,\sigma)}(C'))}), \tag{11}$$

where $\beta$ is a hyperparameter to be discussed later. Eq. 11 assigns a higher probability to clusterings with a smaller clustering cost in the triangle $t$. Then, we can compute the expected cost of a triangle $t$ as

$$\mathbb{E}[\Delta_t] := \mathbb{E}_{C \sim \boldsymbol{P}(\mathcal{C}_t)}[\Delta_{(t,\sigma)}(C)] = \sum_{C \in \mathcal{C}_t} p(C|t) \Delta_{(t,\sigma)}(C). \tag{12}$$

While maxmin gives all good triangles a score of 0, maxexp will not do so in general (except if $\beta = \infty$, see next section). However, a larger expected cost for a good triangle does not imply that it is more informative, like it does for a bad triangle. Therefore, we restrict maxexp to query edge weights from bad triangles, i.e., those that are *guaranteed* to contribute to an inconsistency:

$$\hat{t} = \arg\max_{t \in \mathcal{T}_\sigma} \mathbb{E}[\Delta_t]. \tag{13}$$

We can then select the edge from $\hat{t}$ with the smallest absolute edge weight, i.e., $\hat{e} = \min_{e \in \boldsymbol{E}_i} |\sigma(e)|$. A batch $\mathcal{B} \subseteq \boldsymbol{E}$ is selected as for maxmin (see Section 3.5 for details). Thus, in summary, maxexp queries edges in bad triangles with the largest expected clustering cost. In the case that $\mathcal{T}_{\sigma_i} = \emptyset$ in iteration $i$, we query an edge weight uniformly at random. In Appendix B.2 we discuss some further extensions of maxexp.

### 3.4 Further analysis of maxmin and maxexp

In this section, we analyze the behavior of maxexp compared to maxmin. Any triangle $t = (u, v, w) \in \boldsymbol{T}$ can be clustered in five different ways, i.e., $\mathcal{C}_t = \{C^{(t,1)}, \ldots, C^{(t,5)}\} = \{\{(u,v,w)\}, \{(u),(v,w)\}, \{(v),(u,w)\},$ $\{(w),(u,v)\}, \{(u),(v),(w)\}\}$ (order is important here). The pairwise similarities of a triangle $t \in \boldsymbol{T}$ are denoted by $\{s_1, s_2, s_3\} := \{\sigma(u,v), \sigma(u,w), \sigma(v,w)\}$ (order is important here too). Let $p_j := p(C^{(t,j)}|t)$ for $j = 1, \ldots, 5$. Then, Proposition 1 provides an expression for the expected cost $\mathbb{E}[\Delta_t]$.

**Proposition 1.** *Given a bad triangle $t \in \mathcal{T}_\sigma$ where $s_1$ and $s_2$ are positive and $s_3$ is negative. We have*

$$
\begin{aligned}
\mathbb{E}[\Delta_t] = {} & p_1|s_3| + p_2(|s_1| + |s_2| + |s_3|) + \\
& p_3|s_1| + p_4|s_2| + p_5(|s_1| + |s_2|)
\end{aligned}
\tag{14}
$$

*Proof.* See Appendix A.2. $\qquad\square$

From Proposition 1 we conclude two insights. First, the larger the pairwise similarities are in absolute value the larger the expected cost (i.e., maxexp prioritizes the bad triangles that are more confidently inconsistent). Second, the two positive edges violate three of the five clusterings while the negative edge only violates two of the five clusterings. This implies that there is a bias towards the bad triangles wherein the positive edge weights are larger compared to the negative edge weight in absolute value. See Appendix B.1 for more details.

Furthermore, maxmin can be regarded as a special case of maxexp when $\beta = \infty$, as shown in Proposition 2.

**Proposition 2.** *Given a bad triangle $t \in \mathcal{T}_\sigma$ where $s_1$ and $s_2$ are positive and $s_3$ is negative. The optimization problems in Eq. 10 and Eq. 13 are equivalent if $\beta = \infty$, i.e.,*

$$
\lim_{\beta \to \infty} \mathbb{E}[\Delta_t] = \min_{e \in \boldsymbol{E}_t} |\sigma(e)|, \quad \forall t \in \mathcal{T}_\sigma
\tag{15}
$$

*Furthermore, if $\beta = 0$ we have*

$$
\lim_{\beta \to 0} \mathbb{E}[\Delta_t] \propto |s_1| + |s_2| + \frac{2}{3}|s_3|.
\tag{16}
$$

*Proof.* See Appendix A.3. $\qquad\square$

Proposition 2 implies that $\beta$ can be used to control the level of confidence we put in each of the clusterings when computing the expected cost. The most optimistic case is when $\beta = \infty$ where we put all the confidence in the most likely clustering (i.e., the clustering with the smallest cost). The most pessimistic case is when $\beta = 0$ where all clusterings are weighted equally. Intermediate values of $\beta$ will result in different weightings of the clusterings, leading to different rankings of the triangles. The best value of $\beta$ depends on the problem, and usually, a wide range of values is acceptable.

### 3.5 Efficient implementation of maxmin and maxexp

Solving the optimization problems of maxmin and maxexp (Eq. 10 and Eq. 13, respectively) requires enumerating all the $|\mathcal{T}| = \binom{N}{3} = \mathcal{O}(N^3)$ triangles. However, we are only concerned with the bad triangles $\mathcal{T}_\sigma \subseteq \mathcal{T}$, i.e., the triangles for which there is no clustering $C \in \mathcal{C}_t$ such that $\Delta_{(t,\sigma)}(C) = 0$. This means that at least one edge in each bad triangle $t \in \mathcal{T}_\sigma$ has to violate the current clustering $C^i$. Given this, we maintain a set $\mathbf{E}_i^{\text{violates}}$ of edges that violate the current clustering $C^i$. Constructing this set is $\mathcal{O}(N^2)$, since we need to iterate over all $\binom{N}{2}$ edges. Given $\mathbf{E}_i^{\text{violates}}$, we can efficiently locate all bad triangles without having to iterate over all $\binom{N}{3}$ triangles as follows. We iterate over each edge $(u,v) \in \mathbf{E}_i^{\text{violates}}$, and for each

edge, we iterate over all remaining objects $\mathbf{V} \setminus \{u, v\}$ which gives us all the bad triangles. This step requires $|\mathbf{E}_i^{\text{violates}}| \times N$ iterations, which is $\mathcal{O}(N^3)$ in the worst case when all edges violate $C^i$, i.e., when $\mathbf{E}_i^{\text{violates}} = \mathbf{E}$. However, in practice $|\mathbf{E}_i^{\text{violates}}| \ll |\mathbf{E}|$ which makes this procedure significantly more efficient. Despite this, to further improve the efficiency (as well as the exploration of maxmin and maxexp, as discussed below), we consider a random subset of $\mathbf{E}_i^{\text{violates}}$ of size $N$. The total iterations of the whole procedure will then be $\binom{N}{2} + N^2 = \mathcal{O}(N^2)$, which is a significant improvement over naively iterating over all $\mathcal{O}(N^3)$ triangles. Algorithm 3 outlines our implementation of maxmin and maxexp.

**Exploration with maxmin and maxexp.**  If we have $\mathcal{T}_{\sigma^*} = \emptyset$ and $\mathcal{T}_{\sigma_0} = \emptyset$, then maxmin and maxexp can be interpreted as follows. They begin querying edges randomly (since $\mathcal{T}_{\sigma_0} = \emptyset$ implies that there are no bad triangles initially) and once bad triangles appear due to the new queries, they will begin to actively fix the bad triangles (i.e., turn them into good triangles). If the noise level $\gamma$ is large, they might waste too many queries on fixing bad triangles, leading to a lack of exploration of other edges, thus degrading the performance in the long run. By sampling $N$ edges from $\mathbf{E}_i^{\text{violates}}$ we only consider a random subset of the bad triangles (as explained before), which results in improved exploration overall. We have observed improved performance of this method compared to considering all the bad triangles. In addition, sampling the bad triangles can enhance the diversity of the $B$ edges queried in each iteration which can be useful in batch active learning (Ren et al., 2021).

There are various other options to alleviate the exploration issue. In this paper, we consider two additional methods. First, we consider a hard limit on the number of times one can query an edge. This limit is denoted by $\tau$ and we have $\max_{e \in \mathbf{E}} |\mathcal{Q}_e^i| \le \tau$ for all $i$. Second, we consider $\epsilon$-greedy selection where one queries edge weights according to maxmin (or maxexp) with probability $1 - \epsilon$ or uniformly at random with probability $\epsilon$. More sophisticated exploration techniques adopted from multi-armed bandits can also be employed which we postpone to future work.

---

**Algorithm 3** Efficient implementation of maxmin and maxexp

    **Input**: Data objects $\boldsymbol{V}$, current similarity function $\sigma_i$, current clustering $C^i$, batch size $B$
    **Output**: Batch $\mathcal{B}$
1:   $N \leftarrow |\mathbf{V}|$
2:   $\mathbf{E}_i^{\text{violates}} \leftarrow \{(u, v) \in \mathbf{E} \mid \mathbf{R}_{(\mathbf{V}, \sigma_i, C^i)}(u, v) > 0\}$
3:   $\mathbf{E}_i^{\text{rand}} \leftarrow$ Uniform random subset of $\mathbf{E}_i^{\text{violates}}$ of size $N$
4:   $\mathcal{I} \leftarrow \mathbf{0}^{N \times N}$                                          $\triangleright$ $N \times N$ matrix of zeros.
5:   **for** $(u, v) \in \mathbf{E}_i^{\text{rand}}$ **do**
6:      **for** $w \in \mathbf{V} \setminus \{u, v\}$ **do**
7:          **if** $t = (u, v, w)$ is a bad triangle **then**
8:              $\mathbf{E}_t^{\min} \leftarrow \arg\min_{e \in \mathbf{E}_t} |\sigma_i(e)|$
9:              $e_t^{\min} \leftarrow$ Randomly selected edge from $\mathbf{E}_t^{\min}$
10:             **if** maxmin **then**
11:                 $\mathcal{I}(e_t^{\min}) \leftarrow |\sigma_i(e_t^{\min})|$
12:             **end if**
13:             **if** maxexp **then**
14:                 $\mathcal{I}(e_t^{\min}) \leftarrow \max(\mathbb{E}[\Delta_t], \mathcal{I}(e_t^{\min}))$          $\triangleright$ $\mathbb{E}[\Delta_t]$ as defined in Eq. 12.
15:             **end if**
16:          **end if**
17:      **end for**
18: **end for**
19: $\mathcal{B}^* \leftarrow \arg\max_{\mathcal{B} \subseteq \mathbf{E}, |\mathcal{B}| = B} \sum_{e \in \mathcal{B}} \mathcal{I}(e)$
20: Return $\mathcal{B}^*$

---

## 4   Experiments

In this section, we describe our experimental studies, where additional results are presented in Appendix C.

## 4.1 Experimental setup

**Initial pairwise similarities.** For each experiment, we are given a dataset with ground-truth labels, where the ground-truth labels are only used for evaluations. Then, for each $(u, v) \in \boldsymbol{E}$ in a dataset, we set $\sigma^*(u, v)$ to $+1$ if $u$ and $v$ belong to the same class, and $-1$ otherwise. In active learning, $\sigma^*$ is not available to the correlation clustering algorithm. The similarity function $\sigma_0$ can for example be initialized with random weights or based on a partially correct pre-computed clustering. We investigate both methods in our experiments. For random initialization, we randomly assign each of the objects to one of the ten different clusters resulting in a clustering $C$. Then, for each $(u, v) \in \boldsymbol{E}$, the initial similarity $\sigma_0(u, v)$ is set to $+0.1$ if $u$ and $v$ are in the same cluster according to $C$, and $-0.1$ otherwise. The initial similarities are set to 0.1 in absolute value in order to indicate uncertainty about their correct values. In the other initialization method, we run k-means++ on the corresponding dataset which results in a clustering that is given to the active learning procedure. The clustering is converted into a similarity function in the same way as described above. The feature vectors are used by k-means++ to yield the initial clustering, but the active learning procedure does not have access to them and only uses the initial pairwise similarities.

**Clustering algorithm.** All the experiments use the correlation clustering algorithm $\mathcal{A}$ described in Section 2.4.

**Query strategies.** We consider five different query strategies: **uniform**, **uncertainty** (Eq. 8), **frequency** (Eq. 9), **maxmin** (Eq. 10) and **maxexp** (Eq. 13). We set $\epsilon = 0.3$, $\tau = 5$ and $\beta = 1$ for all experiments unless otherwise specified. See Appendix C for experiments with different values of these parameters.

**Baselines.** As previously discussed, none of the existing methods naturally align with our framework, especially in terms of supporting batch selection of a fixed size. Consequently, we have adapted these methods to fit our framework by performing the corresponding method from scratch while maintaining a fixed query budget equivalent to the total number of edges queried in the current iteration $i$ of our framework. Below, we provide a description of each of the baseline methods.

- **QECC**. This method is a pivot-based active correlation clustering algorithm introduced in (García-Soriano et al., 2020). It starts by randomly selecting a pivot (object) $u \in \boldsymbol{V}$ and then queries the similarity between $u$ and all other objects $\boldsymbol{V} \setminus u$ to form a cluster with $u$ and all the objects with a positive similarity with $u$. It then continues iteratively with the remaining objects until all objects are assigned to a cluster or if the query budget is reached. More specifically, we implement QECC-heur (also introduced in (García-Soriano et al., 2020)) which is a heuristic improvement over QECC, but the general idea of this method is the same as QECC.

- **COBRAS**. This is an active semi-supervised clustering method developed in the constraint clustering setting. It is an extension of **COBRA** (van Craenendonck et al., 2018a). COBRA initially clusters the data into $k$ clusters (super-instances) using $k$-means. The medoid of each cluster is then considered a representative of the super-instance. The goal is to cluster the medoids (and thus indirectly all data points represented by the medoids) with pairwise queries between the medoids. Thus, the pairwise relations involving all non-medoid objects are assumed fixed after running k-means initially, which may be a significant limitation if the feature representation is insufficient. COBRAS extends COBRA by allowing refinement of the super-instances as more pairwise queries are made.

- **nCOBRAS**. This method is also an active constraint clustering method introduced in (Soenen et al., 2021). nCOBRAS extends COBRAS to the noisy setting by attempting to detect and correct the noisy pairwise constraints. This method suffers from various limitations and issues, e.g., it assumes the underlying noise level is known (which is very unrealistic), and the performance (and computation time) is severely affected by larger noise levels and larger query budgets.

It is worth mentioning that COBRAS and nCOBRAS use feature vectors which give them an unfair advantage over our methods. Implementations of COBRAS and nCOBRAS are publicly available and are thus used in

our experiments.[5] Finally, we note that there exist other active semi-supervised clustering methods developed in the constraint clustering setting such as NPU (Xiong et al., 2014) used as a baseline in (Soenen et al., 2021). These methods are not directly relevant to our setting: They do not consider correlation clustering, use feature vectors, and assume zero noise. Moreover, it has been shown that COBRAS and nCOBRAS outperform these methods (Soenen et al., 2021), and, therefore, we do not include them in our experiments.

**Performance evaluation.** In each iteration of the active correlation clustering procedure, we calculate the Adjusted Rand Index (ARI) between the current clustering $C^i$ and the ground truth clustering $C^*_{(\mathbf{V}, \sigma^*)}$. Intuitively, ARI measures how similar the two clusterings are, where a value of 1 indicates they are identical. We report results for additional evaluation metrics in Appendix C. Each active learning procedure is repeated 15 times with different random seeds, where the variance is indicated by a shaded color.

**Datasets.** We perform experiments on ten different datasets, all of which are listed and explained below. For a dataset of $N$ objects, the number of pairwise similarities is $|\mathbf{E}| = (N \times (N - 1))/2$, which implies the huge querying space that active learning needs to deal with. We use a batch size of $B = \lceil |\mathbf{E}|/1000 \rceil$ for all datasets, unless otherwise specified.

1. *Synthetic*: consists of $N = 500$ (and $|\mathbf{E}| = 124750$) normally distributed 10-dimensional data points split evenly into 10 clusters.

2. *20newsgroups*: consists of 18846 newsgroups posts (in the form of text) on 20 topics (clusters). We use a random sample of $N = 1000$ posts (with $|\mathbf{E}| = 499500$). The cluster sizes are 46, 57, 45, 48, 63, 47, 48, 65, 51, 53, 39, 57, 46, 56, 65, 47, 52, 48, 38 and 29. For the $k$-means initialization of $\sigma_0$ we use $k = 20$. We use the `distilbert-base-uncased` transformer model loaded from the Flair Python library (Akbik et al., 2018) in order to embed each of the 1000 documents (data points) into a 768-dimensional latent space, in which $k$-means is performed.

3. *CIFAR10*: consists of 60000 $32 \times 32$ color images in 10 classes, with 6000 images per class. We use a random sample of $N = 1000$ images (with $|\mathbf{E}| = 499500$). The cluster sizes are 91, 96, 107, 89, 99, 113, 96, 93, 112 and 104. For the $k$-means initialization of $\sigma_0$ we use $k = 3$. We use a ResNet18 model (He et al., 2015) trained on the full CIFAR10 dataset in order to embed the 1000 images into a 512-dimensional space, in which $k$-means is performed.

4. *MNIST*: consist of 60000 $28 \times 28$ grayscale images of handwritten digits. We use a sample of $N = 1000$ images (with $|\mathbf{E}| = 499500$). The cluster sizes are 105, 109, 111, 112, 104, 86, 99, 88, 88 and 98. For the $k$-means initialization of $\sigma_0$ we use $k = 3$. We use a simple CNN model trained on the MNIST dataset in order to embed the 1000 images into a 128-dimensional space, in which $k$-means is performed.

5. *Cardiotocography*: includes 2126 fetal cardiotocograms consisting of 22 features and 10 classes. We use a sample of $N = 1000$ data points (with $|\mathbf{E}| = 499500$). The cluster sizes are 180, 275, 27, 35, 31, 148, 114, 62, 28, 100. For the $k$-means initialization of $\sigma_0$ we use $k = 10$.

6. *Ecoli*: a biological dataset on the cellular localization sites of 8 types (clusters) of proteins which includes $N = 336$ samples (with $|\mathbf{E}| = 56280$). The samples are represented by 8 real-valued features. The cluster sizes are 137, 76, 1, 2, 37, 26, 5, and 52. For the $k$-means initialization of $\sigma_0$ we use $k = 10$.

7. *Forest Type Mapping*: a remote sensing dataset of $N = 523$ samples with 27 real-valued attributes collected from forests in Japan and grouped in 4 different forest types (clusters) (with $|\mathbf{E}| = 136503$). The cluster sizes are 168, 84, 86, and 185. For the $k$-means initialization of $\sigma_0$ we use $k = 10$.

8. *Mushrooms*: consists of 8124 samples corresponding to 23 species of gilled mushrooms in the Agaricus and Lepiota Family. Each sample consists of 22 categorical features. We use a sample of $N = 1000$

---

[5]Link to the open-source implementations of COBRAS and nCOBRAS: `https://github.com/jonassoenen/noise_robust_cobras`

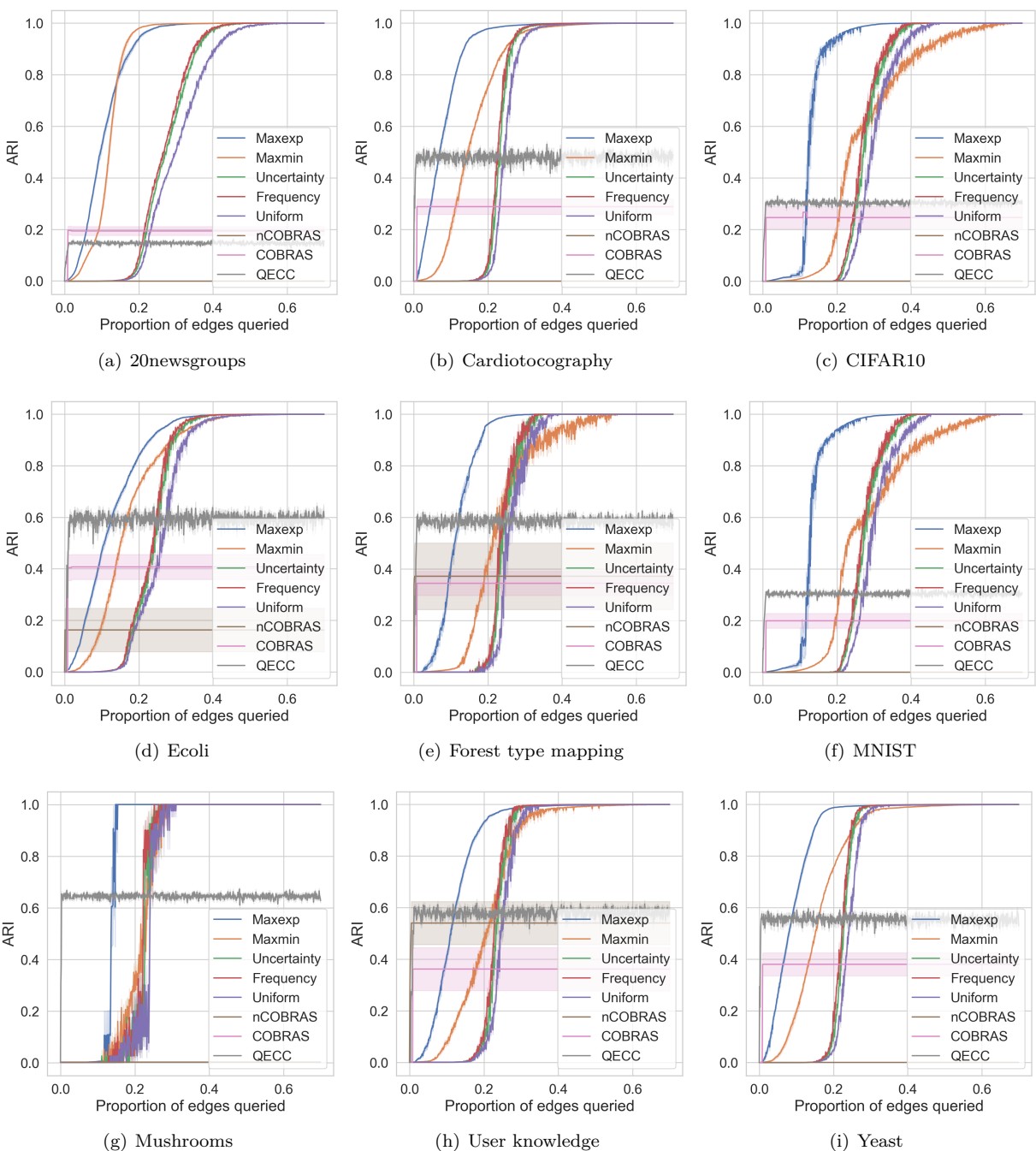

Figure 1: Results for different datasets with 20% noise ($\gamma = 0.2$) and random initialization of the pairwise similarities. The evaluation metric is the adjusted rand index (ARI).

data points (with $|\mathbf{E}| = 499500$). The cluster sizes are 503 and 497. For the $k$-means initialization of $\sigma_0$ we use $k = 2$.

9. *User Knowledge Modelling*: contains 403 students' knowledge status on Electrical DC Machines with 5 integer attributes grouped in 4 categories (with $|\mathbf{E}| = 81003$). The cluster sizes are 111, 129, 116, 28, and 19. For the $k$-means initialization of $\sigma_0$ we use $k = 5$.

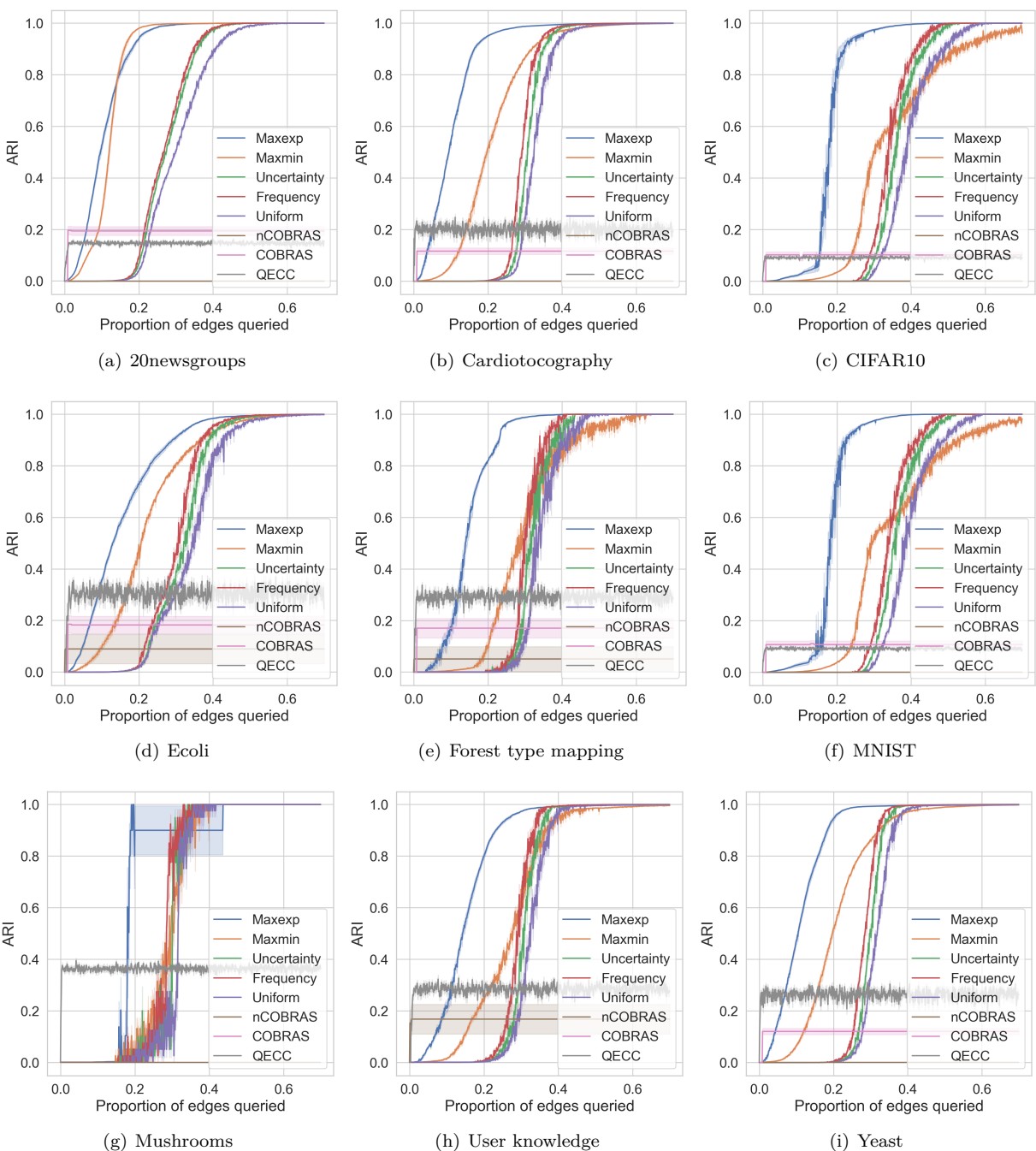

Figure 2: Results for different datasets with 40% noise ($\gamma = 0.4$) and random initialization of the pairwise similarities. The evaluation metric is the adjusted rand index (ARI).

10. *Yeast*: consists of 1484 samples with 8 real-valued features and 10 clusters. We use a sample of $N = 1000$ data points (with $|\mathbf{E}| = 499500$). The cluster sizes are 319, 4, 31, 17, 28, 131, 169, 271, 12 and 18. For the $k$-means initialization of $\sigma_0$ we use $k = 10$.

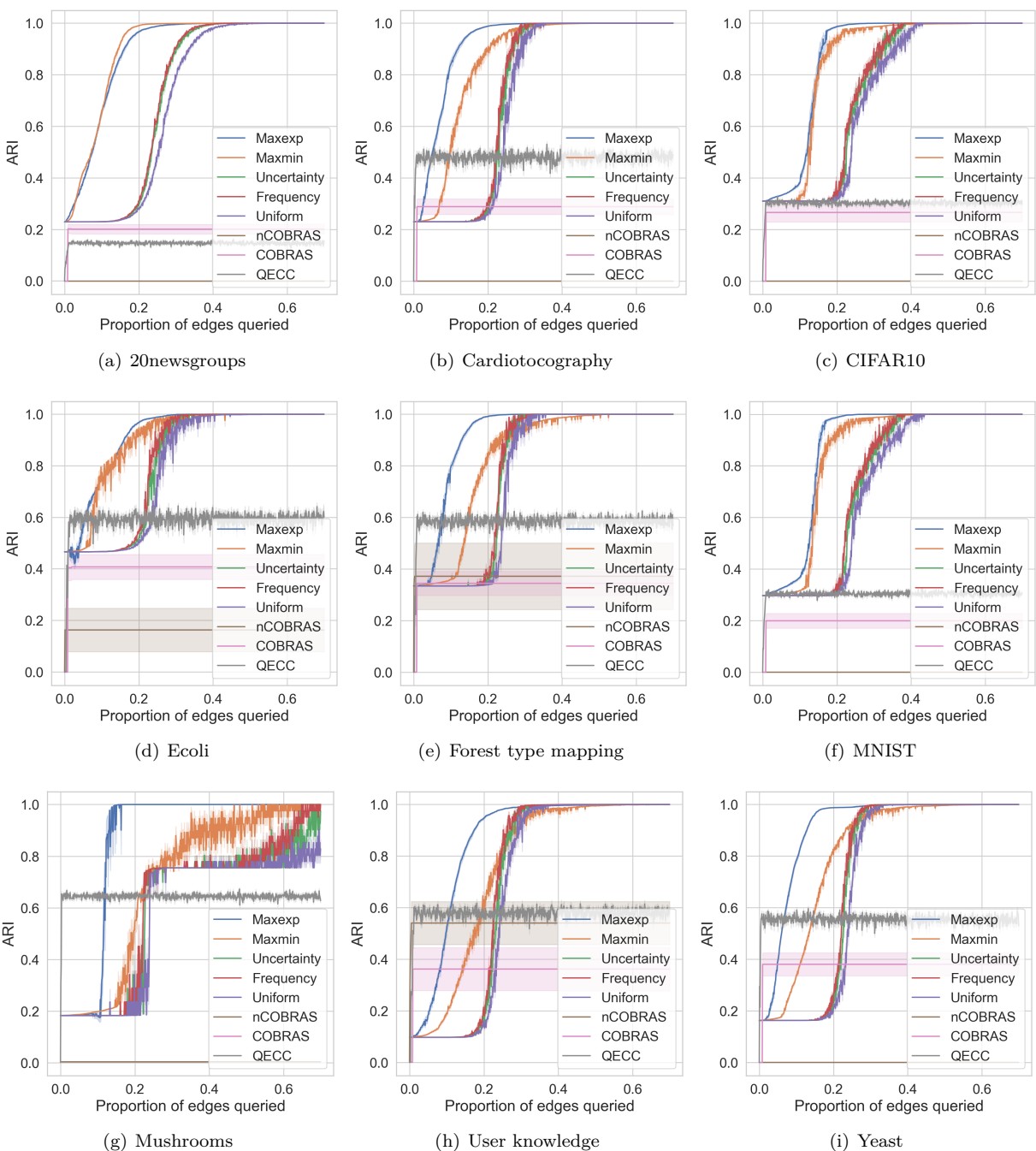

Figure 3: Results for different datasets with 20% noise ($\gamma = 0.2$) and $k$-means initialization of the pairwise similarities. The evaluation metric is the adjusted rand index (ARI).

## 4.2 Experiments on real-world datasets

Figures 1 and 2 illustrate the results for different real-world datasets with a random initialization of $\sigma_0$ and noise levels $\gamma = 0.2$ and $\gamma = 0.4$, respectively. We observe that with even a fairly small amount of noise, all the baselines (nCOBRAS, COBRAS and QECC) perform very poorly. In general, nCOBRAS (which is supposed to be robust to noise) has a lot of trouble with larger query budgets and noise levels. For smaller budgets, nCOBRAS usually outperforms COBRAS in the noisy setting (as demonstrated in (Soenen et al.,

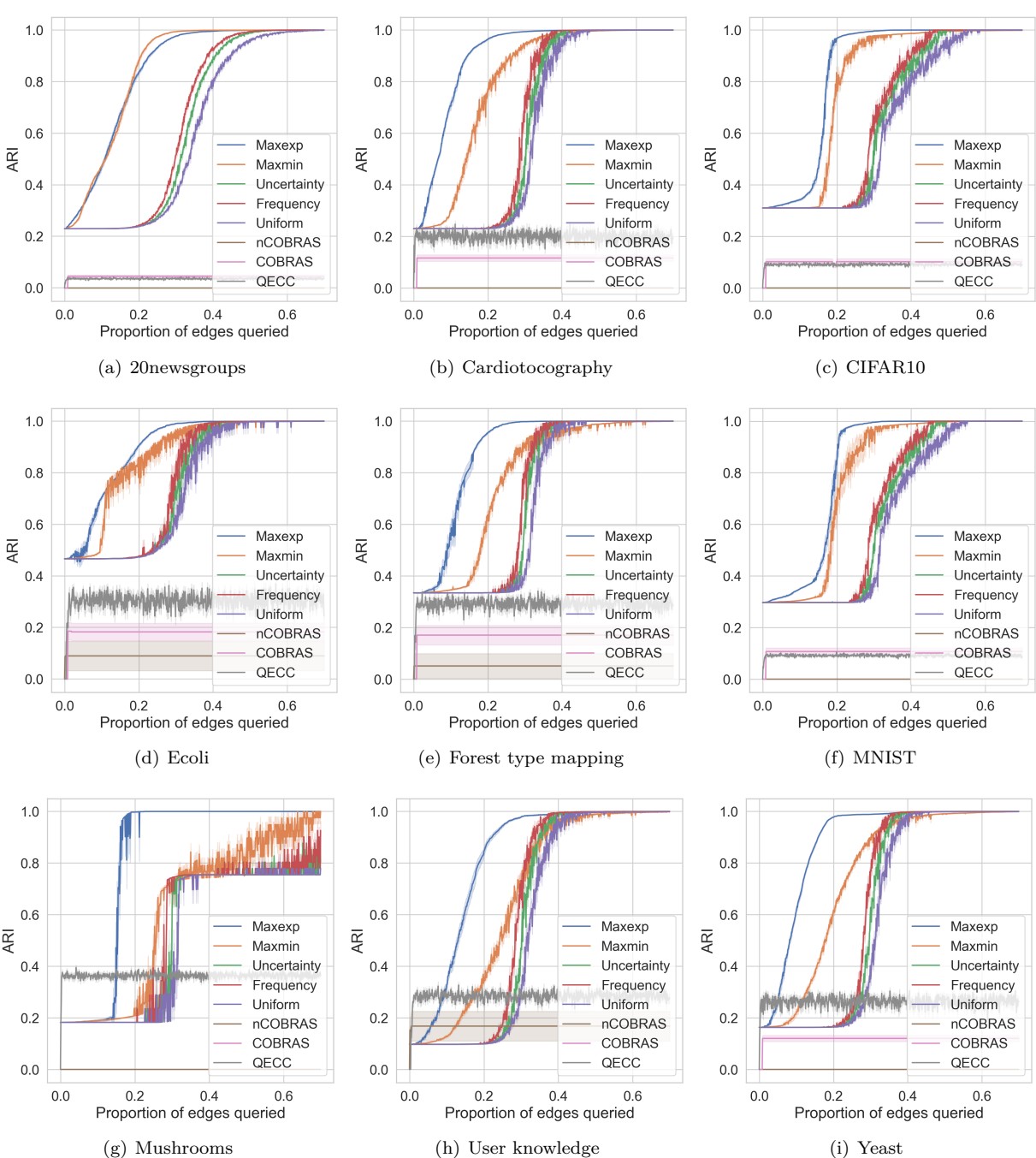

Figure 4: Results for different datasets with 40% noise ($\gamma = 0.4$) and $k$-means initialization of the pairwise similarities. The evaluation metric is the adjusted rand index (ARI).

2021)). However, they tolerate only very small query budgets such as 200, resulting in an ARI of 0.45 on average with 10% noise, which is very low compared to our results. In contrast, our methods can handle any query budget and essentially always converge to optimal clustering (i.e., perfect ARI) even for high noise levels. The inconsistency-based query strategy **maxexp** outperforms the other methods with the best performance overall. We also see that **maxmin** performs well in many cases, but has trouble converging for some of the datasets. As expected, uncertainty and frequency perform very similarly (as discussed in Section 3), while outperforming the random query strategy in all cases.

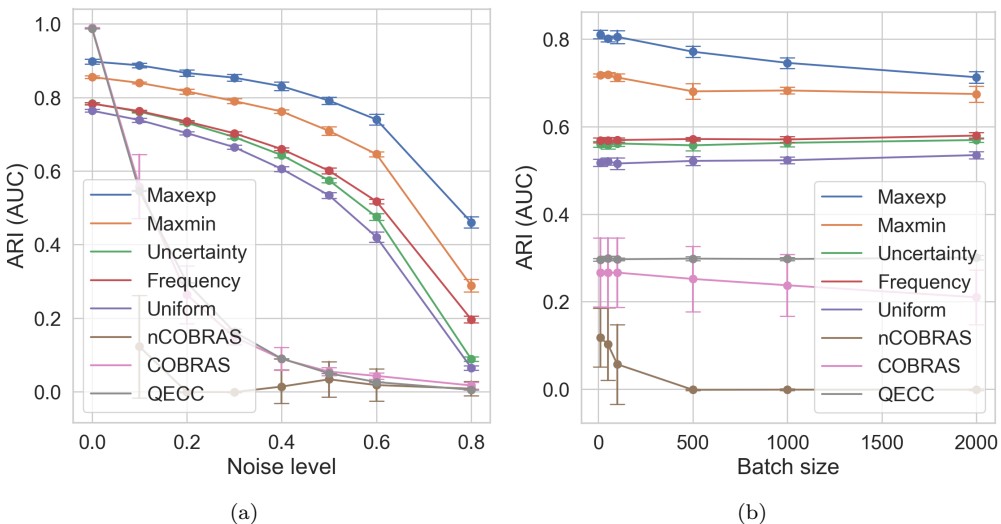

Figure 5: Performance of different query strategies on the synthetic dataset with varying values of the noise level $\gamma$ and batch size $B$. When varying the noise level, we fix $B = \lceil |\mathbf{E}|/1000 \rceil$. When varying the batch size, we fix $\gamma = 0.2$.

Figures 3 and 4 show the results for different real-world datasets with $k$-means initialization of $\sigma_0$ and with noise levels of $\gamma = 0.2$ and $\gamma = 0.4$, respectively. The overall conclusion is very similar to the random initialization of $\sigma_0$, i.e., **maxexp** and then **maxmin** yield the best results with a remarkable margin. We also observe that the performance of our methods is slightly better with this initialization, in particular, **maxmin** seems to perform significantly better on some of the datasets. This is expected since we use some prior information when computing $\sigma_0$.

### 4.3 Analysis of noise level and batch size

Figure 5(a) shows the results for the synthetic dataset when varying the noise level $\gamma$. For each noise level, we keep the batch size fixed at $B = \lceil |\mathbf{E}|/1000 \rceil$. The $y$-axis corresponds to the area under the curve (AUC) of the active learning plot w.r.t. the respective performance metric (i.e., ARI). This metric is used to summarize the overall performance of a query strategy by a single number (instead of the full active learning plots shown in the previous figures). We observe that our methods are very robust to the noise level, whereas the baselines perform well only for zero noise, but very poorly even with a small amount of noise. In particular, we see that **maxexp** is more robust to various noise levels, and **maxmin** is consistently the second best choice.

Finally, Figure 5(b) shows the results for the synthetic dataset when varying the batch size $B$. For each batch size we keep the noise level fixed at $\gamma = 0.2$. The best performing methods are again **maxexp** and **maxmin**, where their performance degrades only slowly as we increase the batch size. We also see that uncertainty and frequency perform similarly for different batch sizes even with small values, where both significantly under perform compared to maxexp and maxmin, indicating that looking at the query frequency or the absolute weight of an edge might not be sufficient. This was already hinted at in Section 3.1, and the reason is that they discard the correlation between edges. In order to make the frequency and uncertainty useful, one needs to combine them with more sophisticated methods that take such correlations into account. This is essentially what maxmin and maxexp do.

### 4.4 Runtime investigation for maxmin and maxexp

Algorithm 3 introduces an efficient implementation of maxmin and maxexp. The key step is line 3 of this algorithm, where we sample a random subset $\mathbf{E}_i^{\mathrm{rand}}$ from the set $\mathbf{E}_i^{\mathrm{violates}}$. In this section, we analyze how

the size of this subset impacts performance and runtime of the active clustering procedure on the synthetic dataset. The noise level is set to $\gamma = 0.4$ for all experiments in this section. Let $\xi$ be a number in $[0, 1]$. Then, let the size of $\mathbf{E}_i^{\text{rand}}$ be $\lfloor \xi \times |\mathbf{E}_i^{\text{violates}}| \rfloor$. Note that $\xi = 1$ means we use $\mathbf{E}_i^{\text{rand}} = \mathbf{E}_i^{\text{violates}}$, which implies that all bad triangles are considered. Also, $\xi = 0$ means no bad triangles are considered, which renders maxmin and maxexp equivalent to uniform selection.

In Figure 6, we investigate the performance and runtime of the maxexp query strategy for different values of $\xi$, in comparison to QECC. Figure 6(a) shows the ARI at each iteration of the active clustering procedure. Figure 6(b) shows the runtime of the query strategy (in seconds) of each iteration. Figure 6(c) shows the number of violations, i.e., $|\mathbf{E}_i^{\text{violates}}|$, at each iteration. Figure 7 illustrates similar results for the maxmin query strategy.

For both maxexp and maxmin, we observe that even very small values of $\xi$ result in very good performance. In fact, smaller values of $\xi$ may even perform better than larger values (in particular for maxmin), due to improved exploration as hinted in Section 3.5. Furthermore, we see that smaller values of $\xi$ result in a significant improvement in the runtime. Setting $\xi = 0.05$ can be seen to perform well for both maxmin and maxexp, while also being comparable with QECC in terms of runtime.

These results also provide insights about the difference between maxmin and maxexp. For example, we see that the number of violations approaches zero for maxexp as the active clustering procedure progresses. Maxmin, on the other hand, does not detect and fix the inconsistencies as effectively. However, maxmin converges to a perfect ARI score slightly faster for this dataset, which indicates that putting too much focus on correcting inconsistencies is not necessarily the best route. In fact, we see that with $\xi = 0$ (i.e., uniform selection), the ARI still reaches an ARI score of 1, despite having a very large number of violations. The conclusion is that while resolving inconsistency in the queried pairwise relations (which maxmin and maxexp aim to do) helps the performance, fixing all of them may not be necessary for optimal performance. This will be investigated in detail in future work.

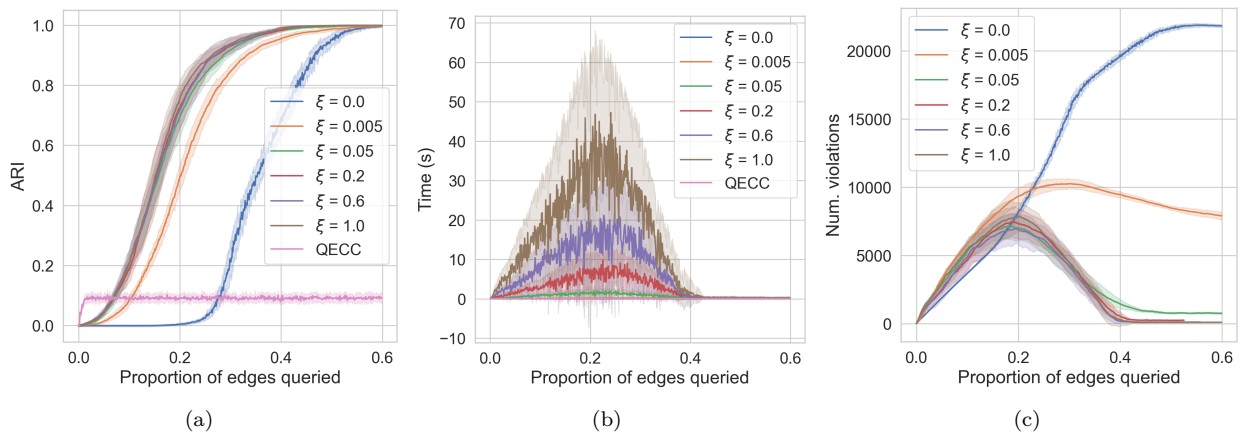

Figure 6: Performance and runtime of the **maxexp** query strategy for different values of $\xi$, in comparison to QECC.

## 5   Conclusion

In this paper, we developed a generic framework for active learning for correlation clustering where queries for pairwise similarities can be any positive or negative real number. It provides robustness and flexibility w.r.t. the type of feedback provided by a user or an annotator. We then proposed a number of novel query strategies suited well to our framework, where maxmin and maxexp yield superior results in various experimental studies.

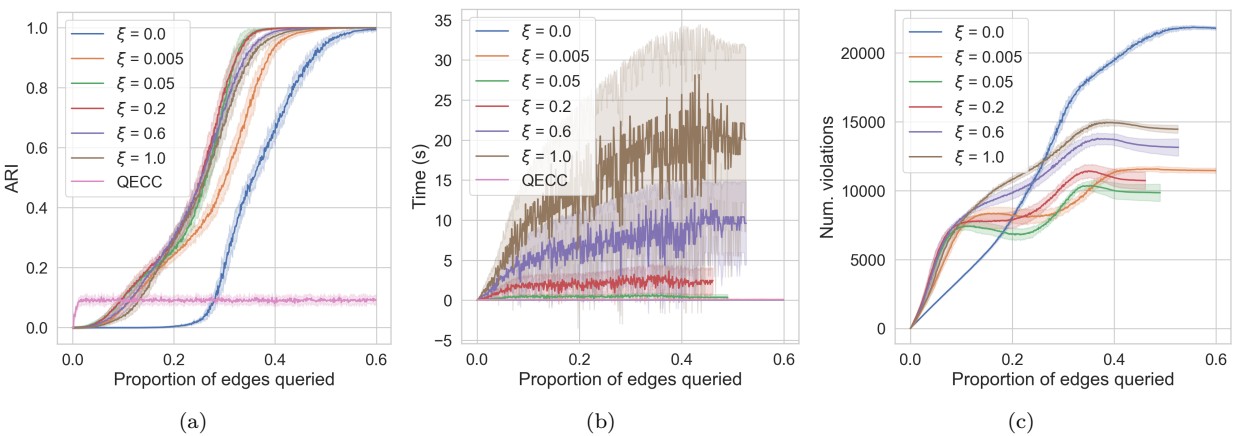

Figure 7: Performance and runtime of the **maxmin** query strategy for different values of $\xi$, in comparison to QECC.

## Acknowledgments

This work was partially supported by the Wallenberg AI, Autonomous Systems and Software Program (WASP) funded by the Knut and Alice Wallenberg Foundation. The computations and data handling were enabled by resources provided by the National Academic Infrastructure for Supercomputing in Sweden (NAISS) and the Swedish National Infrastructure for Computing (SNIC) at Chalmers Centre for Computational Science and Engineering (C3SE), High Performance Computing Center North (HPC2N) and Uppsala Multidisciplinary Center for Advanced Computational Science (UPPMAX) partially funded by the Swedish Research Council through grant agreements no. 2022-06725 and no. 2018-05973.

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

# A    Proofs

## A.1    Proof of theorem 1

**Theorem 1.** *Given $\sigma$, let $\mathcal{T}_\sigma \subseteq \boldsymbol{T}$ be the set of triangles $t = (u, v, w) \in \boldsymbol{T}$ with exactly two positive edge weights and one negative edge weight. Then, the maxmin query strategy corresponds to querying the weight of the edge $\hat{e}$ selected by*

$$(\hat{t}, \hat{e}) = \arg\max_{t \in \mathcal{T}_\sigma} \min_{e \in \boldsymbol{E}_t} |\sigma(e)|. \tag{10}$$

*Proof.* Consider a triangle $t = (u, v, w) \in \boldsymbol{T}$ with pairwise similarities $\{s_1, s_2, s_3\} \coloneqq \{\sigma(u, v), \sigma(u, w), \sigma(v, w)\}$. The clusterings of a triangle $t$ is $\mathcal{C}_t = \{C^1, \ldots, C^5\} = \{(u, v, w), \{(u), (v, w)\}, \{(v), (u, w)\}, \{(w), (u, v)\}, \{(u), (v), (w)\}\}$. Given this, Table 1 shows the clustering cost for each clustering in $\mathcal{C}_t$ for all possible signs of the edge weights $\{s_1, s_2, s_3\}$. From this, we see that only the triangles with two positive edge weights and one negative edge weight have no clustering with a clustering cost of 0. Thus, the triangles in $\mathcal{T}_\sigma$ are guaranteed to be inconsistent while the minimal clustering cost of all other triangles $t \in \boldsymbol{T} \setminus \mathcal{T}_\sigma$ is always 0 (i.e., they can yield a consistent clustering).

Furthermore, for the triangles $t \in \mathcal{T}_\sigma$ we see that the clusterings $C^1$, $C^3$ and $C^4$ corresponds to only one of the edges (i.e., $s_3$, $s_1$ and $s_2$, respectively) violating the clustering. For the clustering $C^2$ all edges violate the clustering and for $C^5$ we have that $s_1$ and $s_2$ violate the clustering. This implies that $C^1$, $C^3$ or $C^4$ will always correspond to the clustering with minimal cost, i.e., $\min_{C \in \mathcal{C}_t} \Delta_{(t, \sigma)}(C) = \min_{e \in \boldsymbol{E}_t} |\sigma(e)|$ for all $t \in \mathcal{T}_\sigma$. Thus, for a given triangle $t \in \mathcal{T}_\sigma$, we should pick the edge according to $\arg\min_{e \in \boldsymbol{E}_t} |\sigma(e)|$.

Finally, among all the triangles $t \in \mathcal{T}_\sigma$, we choose the edge that yields maximal inconsistency, i.e., $(\hat{t}, \hat{e}) = \arg\max_{t \in \mathcal{T}_\sigma} \min_{e \in \boldsymbol{E}_t} |\sigma(e)|$.

$\square$

It should be noted that if $\mathcal{T}_\sigma \neq \emptyset$, then Eq. 10 corresponds to first selecting the triangle $t \in \boldsymbol{T}$ with the maximal minimum clustering cost, i.e.,

$$\hat{t} = \arg\max_{t \in \boldsymbol{T}} \min_{C \in \mathcal{C}_t} \Delta_{(t, \sigma)}(C),$$

and then selecting an edge from $\hat{t}$ according to

$$\hat{e} = \arg\min_{e \in \boldsymbol{E}_{\hat{t}}} |\sigma(e)|.$$

This is true since $\min_{C \in \mathcal{C}_t} \Delta_{(t, \sigma)}(C) \geq 0$ for all $t \in \mathcal{T}_\sigma$ while $\min_{C \in \mathcal{C}_t} \Delta_{(t, \sigma)}(C) = 0$ for all $t \in \boldsymbol{T} \setminus \mathcal{T}_\sigma$.

| $\{s_1, s_2, s_3\}$ | $\Delta_{(t,\sigma)}(C^1)$ | $\Delta_{(t,\sigma)}(C^2)$ | $\Delta_{(t,\sigma)}(C^3)$ | $\Delta_{(t,\sigma)}(C^4)$ | $\Delta_{(t,\sigma)}(C^5)$ |
|---|---|---|---|---|---|
| $\{+, +, +\}$ | $0$ | $|s_1| + |s_2|$ | $|s_1| + |s_3|$ | $|s_2| + |s_3|$ | $|s_1| + |s_2| + |s_3|$ |
| $\{+, +, -\}$ | $|s_3|$ | $|s_1| + |s_2| + |s_3|$ | $|s_1|$ | $|s_2|$ | $|s_1| + |s_2|$ |
| $\{+, -, -\}$ | $|s_2| + |s_3|$ | $|s_1| + |s_3|$ | $|s_1| + |s_2|$ | $0$ | $|s_1|$ |
| $\{-, -, -\}$ | $|s_1| + |s_2| + |s_3|$ | $|s_3|$ | $|s_2|$ | $|s_1|$ | $0$ |

Table 1: Consider a triangle $t = (u, v, w) \in \boldsymbol{T}$ with pairwise similarities $\{s_1, s_2, s_3\} \coloneqq \{\sigma(u, v), \sigma(u, w), \sigma(v, w)\}$. The clusterings of a triangle $t$ is $\mathcal{C}_t = \{C^1, \ldots, C^5\} = \{(u, v, w), \{(u), (v, w)\}, \{(v), (u, w)\}, \{(w), (u, v)\}, \{(u), (v), (w)\}\}$. The first column indicates the sign of the pairwise similarities $\{s_1, s_2, s_3\}$. The remaining columns show the clustering cost for each of the clusterings.

### A.2   Proof of proposition 1

**Proposition 1.** *Given a bad triangle $t \in \mathcal{T}_\sigma$ where $s_1$ and $s_2$ are positive and $s_3$ is negative. We have*

$$
\begin{aligned}
\mathbb{E}[\Delta_t] = p_1|s_3| + p_2(|s_1| + |s_2| + |s_3|) + \\
p_3|s_1| + p_4|s_2| + p_5(|s_1| + |s_2|)
\end{aligned}
\tag{14}
$$

*Proof.* Given the clusterings $\mathcal{C}_t$ it follows directly from the definition of the clustering cost in Eq. 2 in the main text, i.e.,

$$
\begin{aligned}
\mathbb{E}[\Delta_t] &= \sum_{i=1}^{5} p_i \Delta_{(t,\sigma)}(C^{(t,i)}) \\
&= \sum_{i=1}^{5} p_i \sum_{(u,v) \in \boldsymbol{E}_t} \boldsymbol{R}_{(t,\sigma,C^{(t,i)})}(u,v) \\
&= p_1|s_3| + p_2(|s_1| + |s_2| + |s_3|) + \\
&\quad p_3|s_1| + p_4|s_2| + p_5(|s_1| + |s_2|).
\end{aligned}
\tag{17}
$$

The last equality can easily be understood from inspecting the row where $\{s_1, s_2, s_3\} = \{+, +, -\}$ in Table 1. $\square$

### A.3   Proof of proposition 2

**Proposition 2.** *Given a bad triangle $t \in \mathcal{T}_\sigma$ where $s_1$ and $s_2$ are positive and $s_3$ is negative. The optimization problems in Eq. 10 and Eq. 13 are equivalent if $\beta = \infty$, i.e.,*

$$
\lim_{\beta \to \infty} \mathbb{E}[\Delta_t] = \min_{e \in \boldsymbol{E}_t} |\sigma(e)|, \quad \forall t \in \mathcal{T}_\sigma
\tag{15}
$$

*Furthermore, if $\beta = 0$ we have*

$$
\lim_{\beta \to 0} \mathbb{E}[\Delta_t] \propto |s_1| + |s_2| + \frac{2}{3}|s_3|.
\tag{16}
$$

*Proof.* Based on Eq. 11 we see that as $\beta \to \infty$ all probability mass will be put on the clustering(s) with the smallest clustering cost which yields Eq. 15 (from Theorem 1). Furthermore, from Eq. 11 we have that $\lim_{\beta \to 0} p_i = \frac{1}{5}$ for all $i = 1, \dots, 5$ which gives

$$
\begin{aligned}
\lim_{\beta \to 0} \mathbb{E}[\Delta_t] &= \frac{1}{5} \sum_{i=1}^{5} \Delta_{C^{(t,i)}}(\sigma_i) \\
&= \frac{1}{5}(3|s_1| + 3|s_2| + 2|s_3|) \\
&\propto |s_1| + |s_2| + \frac{2}{3}|s_3|.
\end{aligned}
\tag{18}
$$

$\square$

## B   More on Maxmin and Maxexp

In this section, we further investigate maxmin and maxexp, and then discuss some extensions to maxexp.

### B.1 Further analysis of maxmin and maxexp

Figure 8 visualizes how maxexp ranks triangles for different values of $\beta$. We observe that a higher value of $\beta$ makes the maxexp cost more concentrated around bad triangles with the edge weights $\{1, 1, -1\}$. When $\beta < \infty$, maxexp will still give the highest cost to $\{1, 1, -1\}$ (as desired), but the cost for surrounding triangles is assigned in a smoother and more principled manner. Also, while the ranking of good triangles is not relevant in our case, we can see how different values of $\beta$ rank them differently. Additionally, Table 2 illustrates some more subtle differences of maxexp with different values of $\beta$. The main takeaway is that when $\beta < \infty$, the ranking of various triangles follows more accurately the intuition that triangles that are more confidently inconsistent should be ranked higher (i.e., more likely to be selected).

| $\sigma_i(u,v)$ | $\sigma_i(u,w)$ | $\sigma_i(v,w)$ | $\mathbb{E}[\Delta_t\|\beta=1]$ | $\mathbb{E}[\Delta_t\|\beta=\infty]$ | $\mathbb{E}[\Delta_t\|\beta=0]$ |
|---|---|---|---|---|---|
| 1 | 1 | -1 | 1.18 | 1 | 1.6 |
| 0.8 | 0.5 | -0.5 | 0.77 | 0.5 | 0.98 |
| -0.8 | 0.5 | 0.5 | 0.74 | 0.5 | 0.92 |
| 1 | 1 | -0.1 | 0.71 | 0.1 | 1.24 |
| -1 | 1 | 0.1 | 0.69 | 0.1 | 1.06 |
| 1 | 1 | 1 | 0.66 | 0 | 1.8 |
| 0.1 | 0.1 | -0.1 | 0.15 | 0.1 | 0.16 |

Table 2: The expected cost $\mathbb{E}[\Delta_t]$ of a triangle $t = (u, v, w)$ for different values of $\beta$.

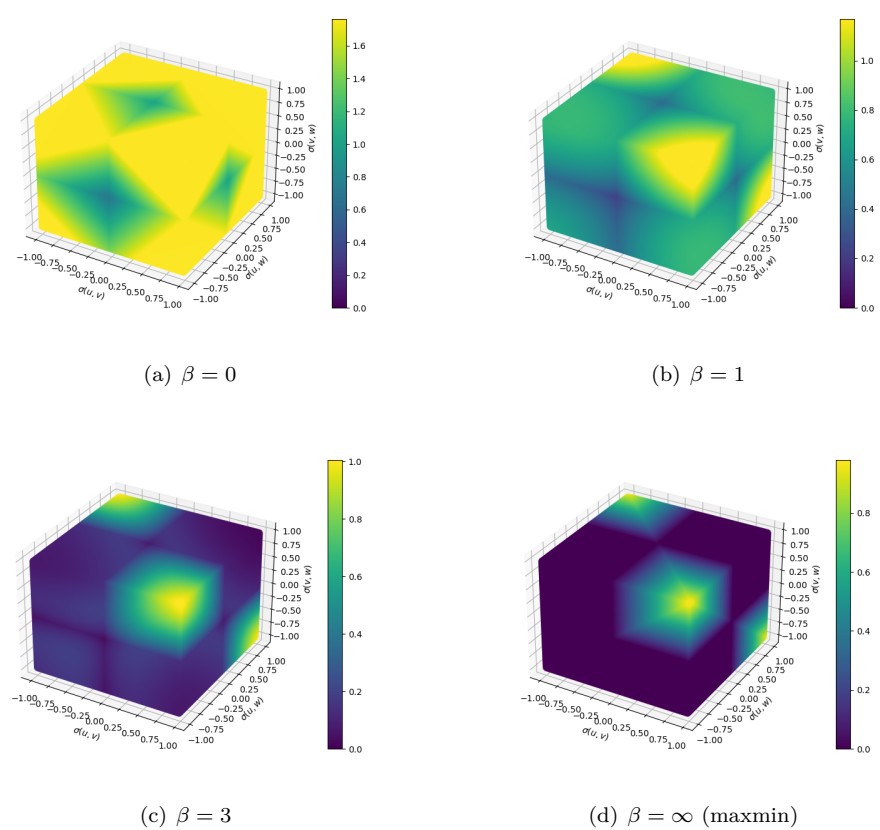

(a) $\beta = 0$

(b) $\beta = 1$

(c) $\beta = 3$

(d) $\beta = \infty$ (maxmin)

Figure 8: The expected cost $\mathbb{E}[\Delta_t]$ of a triangle $t = (u, v, w)$ with different values of $\beta$.

### B.2  Further extensions of maxexp

One can in principle extend the probabilistic perspective to selecting edges and triangles. We can define the probability of an edge $e \in \boldsymbol{E}_t$ given a clustering $C \in \mathcal{C}_t$ as

$$p(e|C) = \frac{\exp(\beta \boldsymbol{R}_{(t,\sigma,C)}(e))}{\sum_{e' \in \boldsymbol{E}_t} \exp(\beta \boldsymbol{R}_{(t,\sigma,C)}(e'))}) \tag{19}$$

Then, after selecting a triangle $t$ using Eq. 13, we first sample a clustering from Eq. 11 and then given this sampled clustering, we sample an edge from Eq. 19. This may result in improved exploration.

Alternatively, we can define the expected cost of an edge $e$ given a triangle $t$ as

$$
\begin{aligned}
\mathbb{E}[\Delta_e|t] &:= \mathbb{E}_{C \sim \boldsymbol{P}(\mathcal{C}_t)}[\boldsymbol{R}_{(t,\sigma,C)}(e)|t] \\
&= \sum_{C \in \mathcal{C}_t} p(C|t)\boldsymbol{R}_{(t,\sigma,C)}(e) .
\end{aligned}
\tag{20}
$$

Then, we define the probability of an edge $e$ given a triangle $t$ as

$$p(e|t) = \frac{\exp(\beta \mathbb{E}[\Delta_e|t])}{\sum_{e' \in \boldsymbol{E}_t} \exp(\beta \mathbb{E}[\Delta_{e'}|t])} . \tag{21}$$

An edge can either be selected according to

$$\hat{e} = \arg\max_{e \in \boldsymbol{E}_t} \mathbb{E}[\Delta_e|t] , \tag{22}$$

or we can sample an edge $\hat{e}$ from $p(e|\hat{t})$.

Additionally, we can define the probability of a triangle given an edge according to

$$p(t|e) = \frac{\exp(\beta \mathbb{E}[\Delta_t])}{\sum_{t' \in \boldsymbol{T}_e} \exp(\beta \mathbb{E}[\Delta_{t'}])} \tag{23}$$

where $\boldsymbol{T}_e \subseteq \boldsymbol{T}$ is the set of triangles containing the edge $e$. Now we can define an informativeness measure of any edge $e \in \boldsymbol{E}$ by

$$
\begin{aligned}
\mathcal{I}(e) &:= \mathbb{E}_{t,C \sim \boldsymbol{P}(\boldsymbol{T}_e, \mathcal{C}_t)}[\boldsymbol{R}_{(t,\sigma,C)}(e)] \\
&= \mathbb{E}_{t \sim \boldsymbol{P}(\boldsymbol{T}_e)}[\mathbb{E}_{C \sim \boldsymbol{P}(\mathcal{C}_t)}[\boldsymbol{R}_{(t,\sigma,C)}(e)|t]] \\
&= \sum_{t \in \boldsymbol{T}_e} p(t|e) \sum_{C \in \mathcal{C}_t} p(C)\boldsymbol{R}_e^{(C,\sigma_i)}
\end{aligned}
\tag{24}
$$

Given this, one could select an edge to be queried according to

$$\hat{e} = \arg\max_{e \in \boldsymbol{E}} \mathcal{I}(e). \tag{25}$$

## C  Experiments: Further Results

In this section, we present several additional experimental results. The experimental setting is identical to the setting in Section 4.1 unless otherwise specified.

## C.1 Results with the adjusted mutual information metric

The results presented in Figures 9-12 are identical to the results shown in Figures 1-4 of the main paper, except with the adjusted mutual information (AMI) as the performance metric. The overall conclusions are very similar to the results with ARI. The main difference is that the performance of our query strategies seems to be somewhat worse for the datasets with unbalanced clusters (e.g., the ecoli dataset), as such datasets make the problem harder.

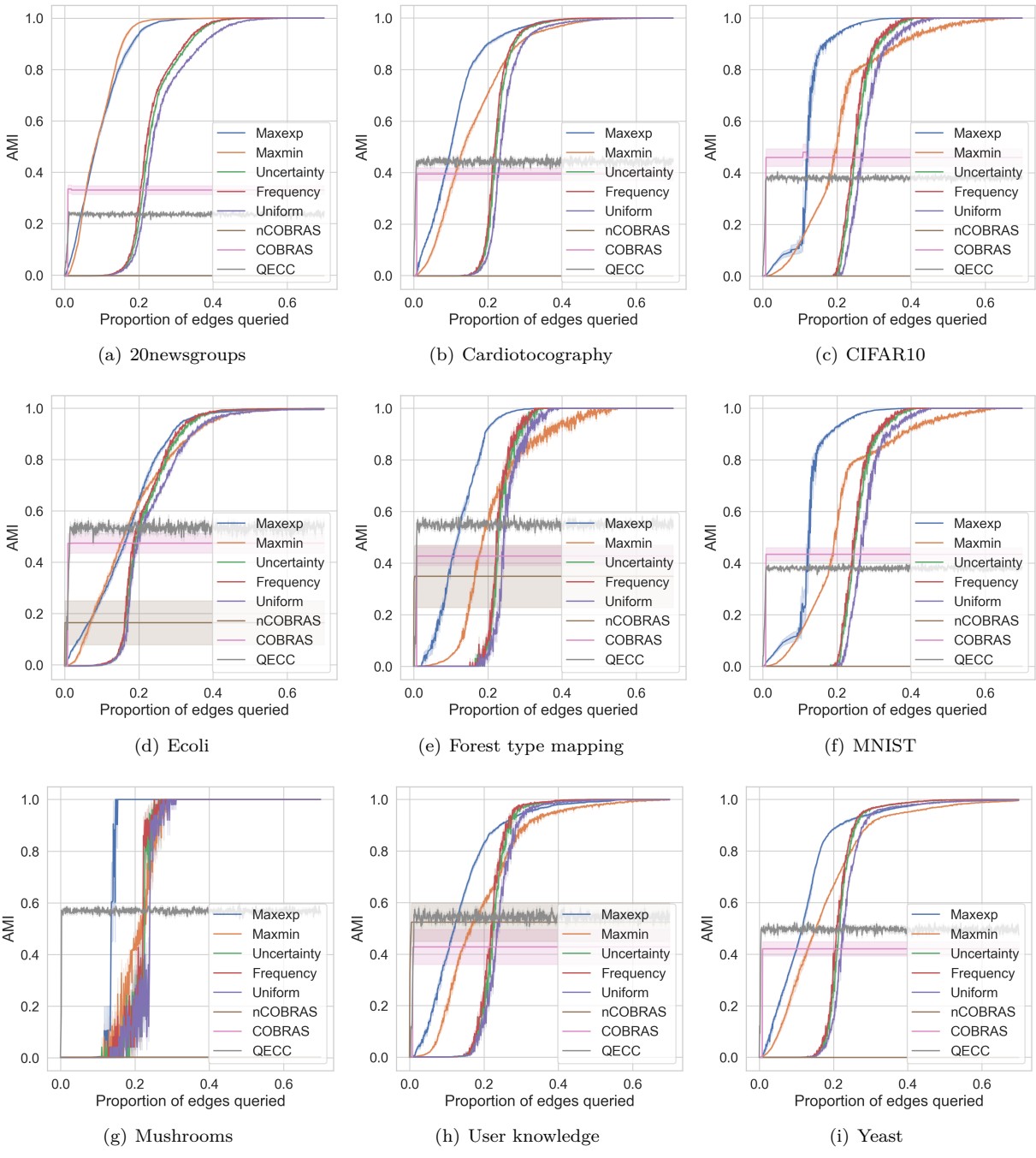

Figure 9: Results for different datasets with 20% noise ($\gamma = 0.2$) and random initialization of the pairwise similarities. The evaluation metric is the adjusted mutual information (AMI).

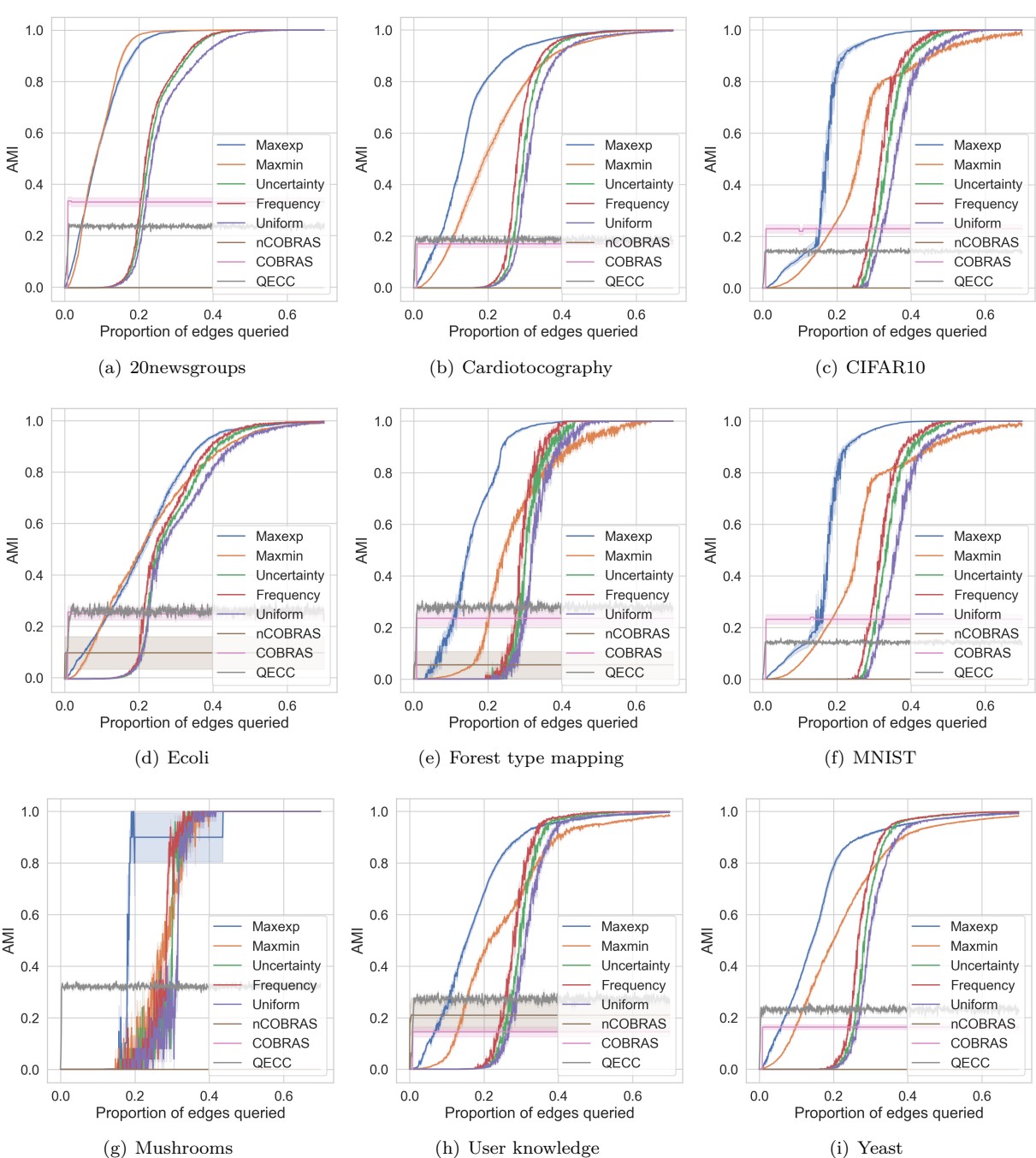

Figure 10: Results for different datasets with 40% noise ($\gamma = 0.4$) and random initialization of the pairwise similarities. The evaluation metric is the adjusted mutual information (AMI).

## C.2   Analysis of the impact of $\beta$ and $\tau$

Figure 13 illustrates the results for the synthetic dataset w.r.t. the adjusted rand index where we investigate the impact of different parameters. Each row shows the results for the noise levels $\gamma = 0, 0.2, 0.4$, respectively. The first column shows the results for all query strategies. The second column shows the results for maxmin and maxexp with $\epsilon = 0, 0.2, 0.4$. The third column shows the results for maxexp with $\tau = 1, 2, 3, 5, 20, \infty$. The fourth column shows the results for maxexp with $\beta = 0, 0.5, 1, 2, 5, 25, \infty$.

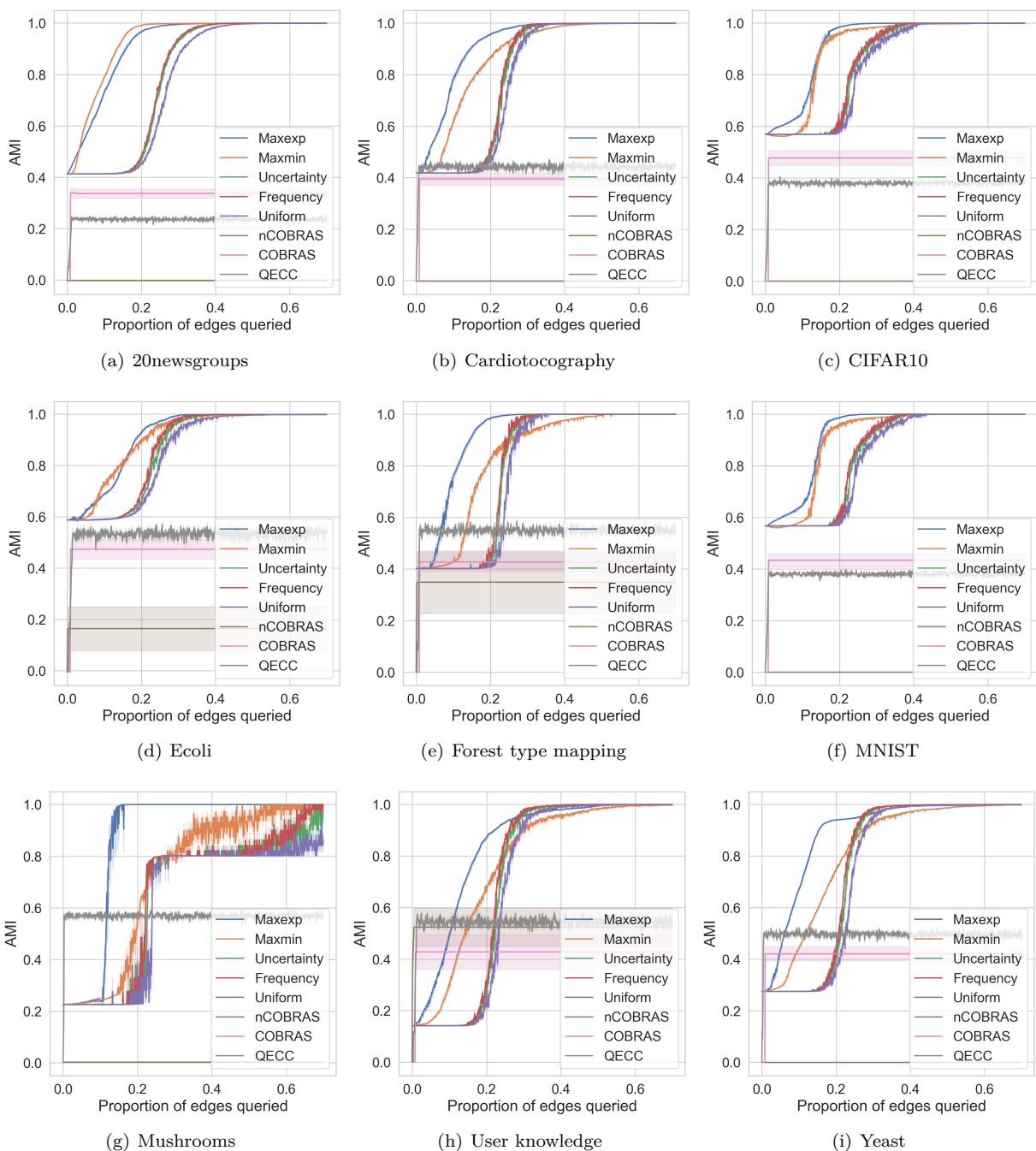

Figure 11: Results for different datasets with 20% noise ($\gamma = 0.2$) and $k$-means initialization of the pairwise similarities. The evaluation metric is the adjusted mutual information (AMI).

From these figures, we conclude consistent results with the previous results. In particular, we observe that for all datasets the performance of maxexp remains the same for various values of $\tau \geq 1$ when $\gamma = 0$ (see, e.g., Figure 13(c)). This makes sense since with zero noise one gains no extra information from querying the same edge more than once, which maxexp (and maxmin) accurately detect. However, for $\gamma = 0.2$, we see (in, e.g., Figure 13(g)) that maxexp performs worse when $\tau < 5$. This confirms that maxexp (and maxmin) benefit from querying the same edge more than once when the oracle is noisy. Finally, when $\gamma = 0.4$, the noise is so high such that maxexp will query the same edges more than what should be, leading to exploration issues.

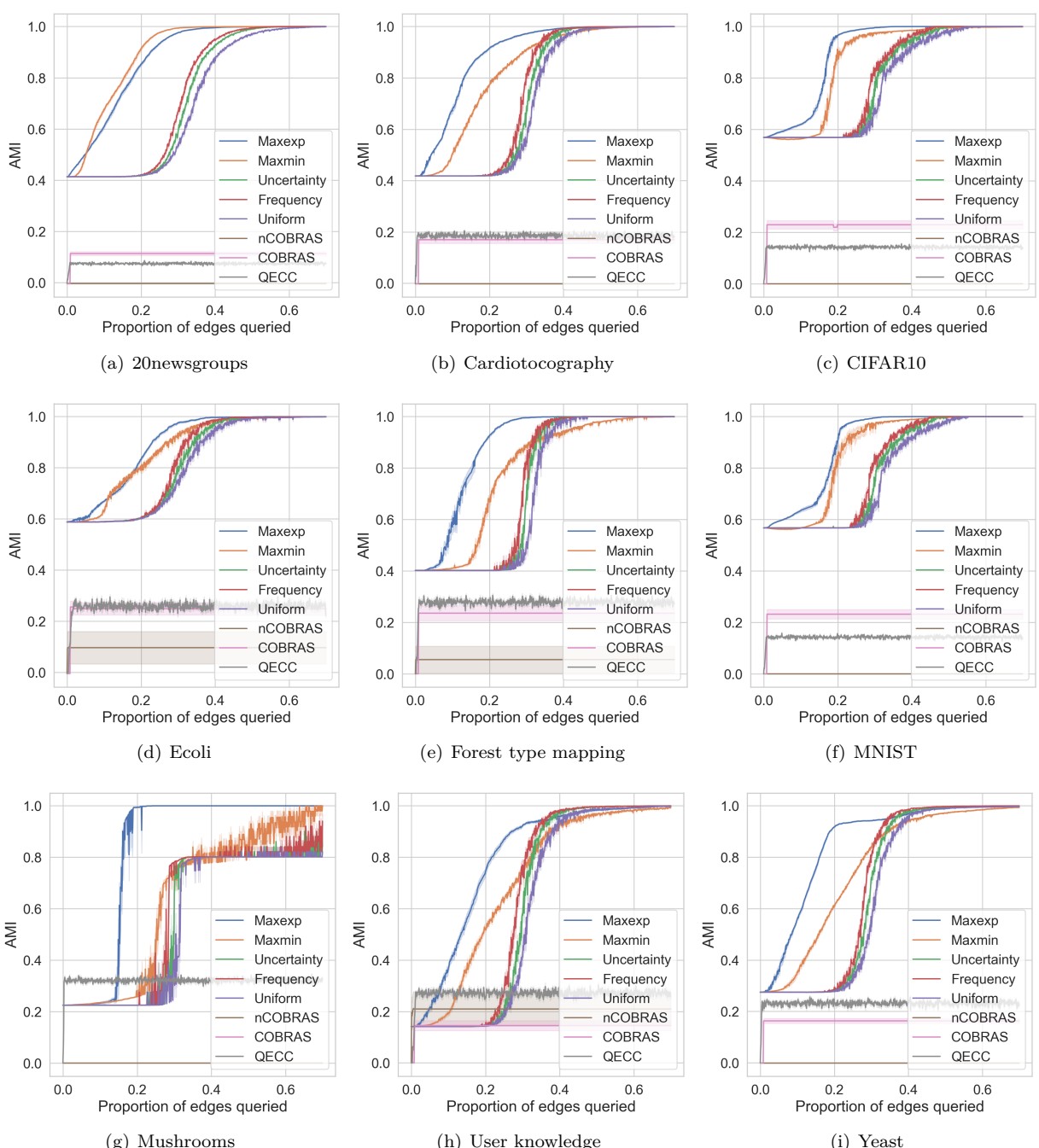

Figure 12: Results for different datasets with 40% noise ($\gamma = 0.4$) and $k$-means initialization of the pairwise similarities. The evaluation metric is the adjusted mutual information (AMI).

This can be observed in, e.g., Figure 13(k) where the performance gets worse when $\tau = \infty$ while the best performance is obtained with $\tau = 20$. Thus, for higher noise levels one cannot simply set $\tau = \infty$ and should limit its value.

From the experiments with different values of $\beta$, we observe that a wide range of values for $\beta$ are acceptable. However, as $\beta \to \infty$ the performance seems to degrade, which in this case maxexp converges to maxmin according to Proposition 2.

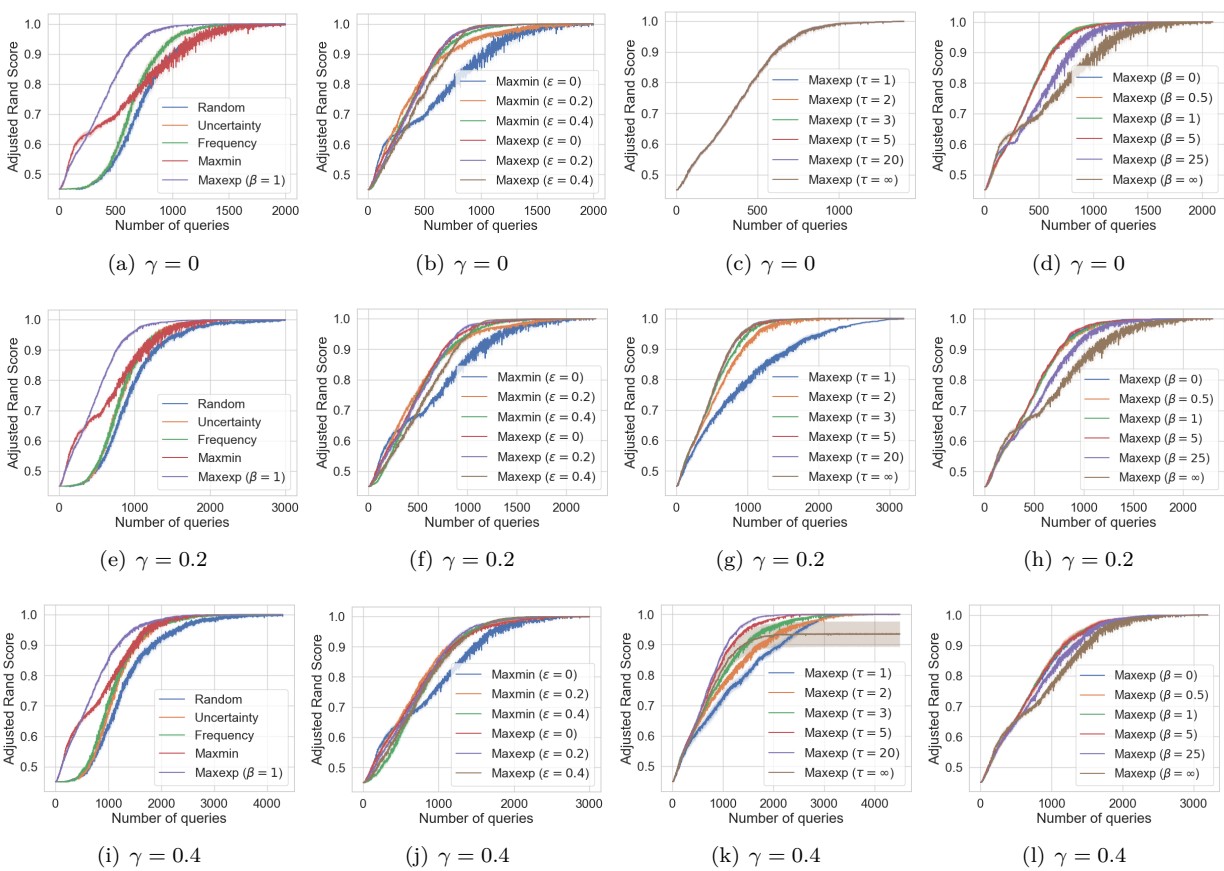

Figure 13: Results for the synthetic dataset in different settings w.r.t. adjusted rand index (ARI). In these experiments, we investigate the impact of different parameters and demonstrate that the results are consistent among different choices of parameters.

## C.3 Analysis of the performance on the MNIST dataset

Figure 14 provides some additional insights into the active clustering procedure applied to the MNIST dataset (with the maxexp query strategy). It shows the computed clusters and the distribution of the true clusters (digits) into these clusters at 11 different iterations. We observe that the quality of the clustering improves as more pairwise relations are queried. It is known that separating digits 1 and 7 tends to be difficult for the MNIST dataset, since they look similar in feature space. However, we see that the procedure does not struggle with any of such cases. The reason for this is that we do not consider the feature vectors, thereby any ambiguity that may exist in the feature space does not affect the performance of our procedure. This shows one benefit of not considering the feature vectors, as they may degrade performance due to the mismatch with the true clustering. Another benefit of not relying on feature vectors is that for certain applications, one may not even have access to features at all, because acquiring them can be costly.

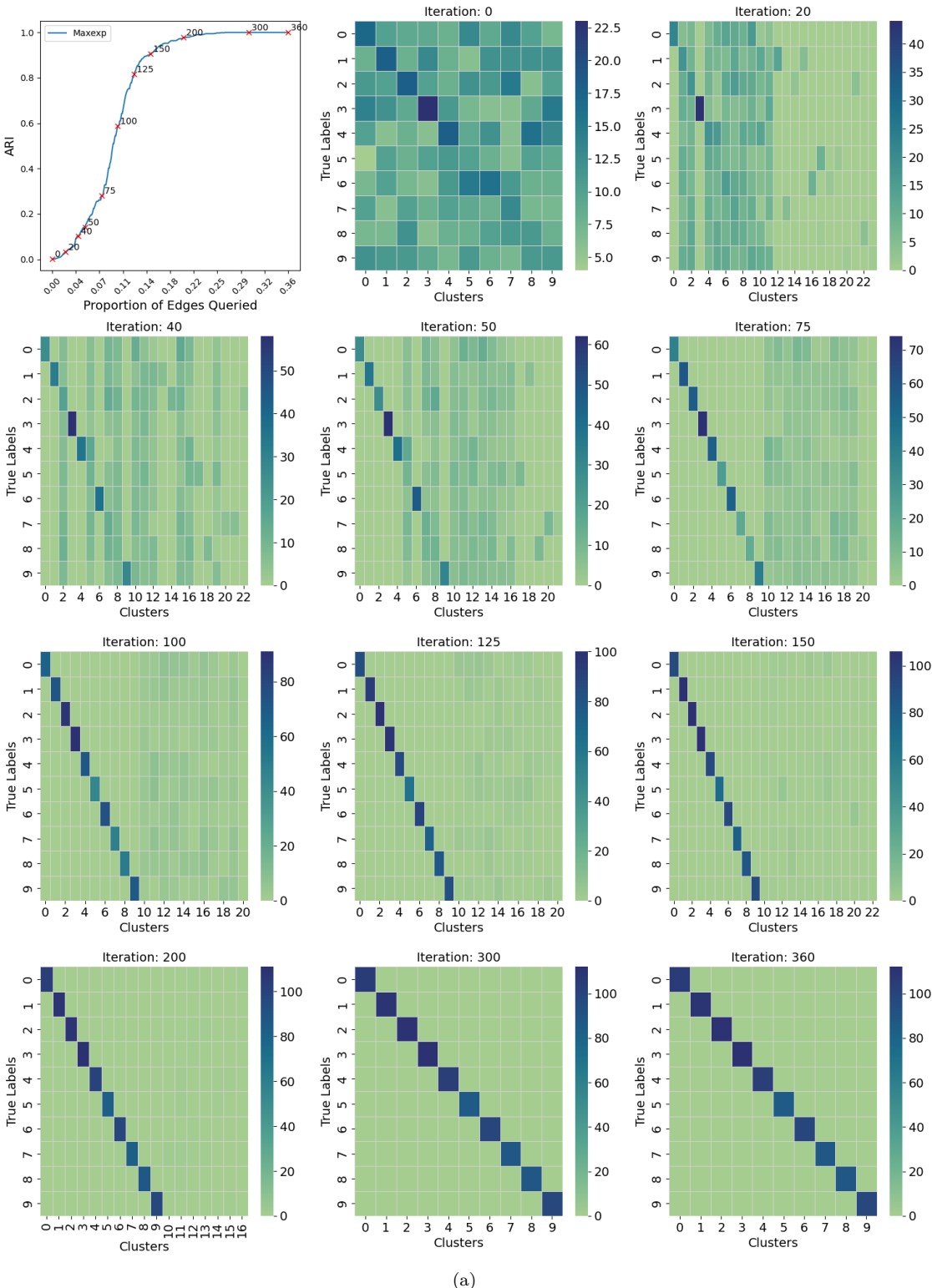

(a)

Figure 14: Performance of the active clustering procedure on the MNIST dataset (with the maxexp query strategy). The clusters found and the distribution of the true labels (digits) into these clusters at 11 different iterations are shown.

