# OpenReview forum: "Correlation Clustering with Active Learning of Pairwise Similarities"
_TMLR — Accepted by TMLR_

### Review · Reviewer_9mYY · 2023-11-14

**Summary Of Contributions:**

The authors propose a novel active learning approach to correlation clustering. Different strategies for querying similarities between data points are considered. The most effective strategy queries the similarities between nodes that are in a "confidently inconsistent" ternary (i.e., triangle) similarity relationship  (e.g., two nodes are close/similar to a third node but dissimilar to each other). Experiments on various datasets and in comparison with SOTA approaches demonstrate the effectiveness of the proposed active clustering technique.

**Audience:**

Yes

**Broader Impact Concerns:**

--

**Claims And Evidence:**

Yes

**Requested Changes:**

1. Please go carefully through the text, as there are some inconsistencies, e.g., on Page 7 "where the first term (i.e., γ) is independent of u and only the second term γ depends on u.".
2. An efficiency and scalability evaluation would considerably improve the work.
3. Are there other, maybe more principled ways to define "confident inconsistency"? A discussion might be insightful and better motivate your approach.

**Strengths And Weaknesses:**

Strengths:

1. The paper is well-written.
2. The approach is well-grounded in theoretical concepts like the transitivity of similarities, clustering cost and adaptive number of clusters for correlation clustering.
3. The approach is extensively evaluated on various datasets and in comparison with SOTA approaches to active correlation clustering.

Weaknesses:
1. Querying similarities of data points in triangle relationships can be quite expensive, and although the authors employ a smart/efficient querying strategy, an efficiency comparison on different datasets and with different SOTA approaches might give further insights on the practical scalability of the approach.
2. Along similar lines, it is unclear how the approach would behave in realistic dynamic scenarios, where data may change over time or new data points have to be considered.
3. Qualitative experiments, e.g., on datasets like MNIST (for example, similar digits such as 1,7, should be grouped into the same cluster), could further improve the presentation.

---

> ### Author Response · Authors · 2023-12-07
> **Response to reviewer 9mYY**
>
> Thank you for the review and the helpful comments! We have attempted to address them in the revised version and in our responses below.
>
> **"Querying similarities of data points in triangle relationships can be quite expensive, and although the authors employ a smart/efficient querying strategy, an efficiency comparison on different datasets and with different SOTA approaches might give further insights on the practical scalability of the approach."**
>
> We have now added Section 4.4 to the revised version, which analyzes the runtime/performace of maxexp and maxmin.
>
> **"Along similar lines, it is unclear how the approach would behave in realistic dynamic scenarios, where data may change over time or new data points have to be considered."**
>
> In this work, we study the pool-based active learning setting where we assume there is a pool of unlabeled data (i.e., the pairwise similarities whose values are not known perfectly in advance) and the goal is to acquire the most informative data (pairwise similarities) in a cost-efficient way. Stream-based active learning is an alternative setting which deals with streaming data (where new data is arriving over time and its distribution might change). We postpone this interesting setting to future work.
>
> **"Qualitative experiments, e.g., on datasets like MNIST (for example, similar digits such as 1,7, should be grouped into the same cluster), could further improve the presentation."**
>
> We have added Section C.3 to the appendix of the revised version, which includes new experiments that address this comment. We note that we do not consider the feature vectors, which means that any ambiguity that may exist in the feature space does not affect the performance of our procedure. Another benefit of not relying on feature vectors is that for certain applications, one may not even have access to features at all, because obtaining them might be costly.
>
> **"Please go carefully through the text, as there are some inconsistencies, e.g., on Page 7 "where the first term (i.e., γ) is independent of u and only the second term γ depends on u."**.
>
> This has now been fixed in the revised version.
>
> **"An efficiency and scalability evaluation would considerably improve the work."**
>
> We have now added Section 4.4 to the revised version, which analyzes the runtime of maxexp and maxmin. We also would like to add that in active learning, usually the key objective is minimizing the total number of queries (or the respective querying cost) rather than the computational aspects. Therefore, in this work, we mainly focus on the querying cost. Finally, it is notable that the runtime of the proposed querying strategies in O(N^2), i.e., comparable to the baseline. This is in particular noticeable when we consider that the cardinality of the querying space (i.e., the size of the space from which we can choose an element to be queried) is also O(N^2).
>
> **"Are there other, maybe more principled ways to define "confident inconsistency"? A discussion might be insightful and better motivate your approach."**
>
> In the context of our work, inconsistency refers to violation of transitivity (transitivity means that that if $\sigma(u, v) \geq 0$ and $\sigma(u, w) \geq 0$ then $\sigma(v, w) \geq 0$ or if $\sigma(u, v) \geq 0$ and $\sigma(u, w) < 0$ then $\sigma(v, w) < 0$.).
> Whenever we say a triangle $t_1$ is "more confidently inconsistent" compared to triangle $t_2$, we mean that $t_1$ (i.e., the objects in $t_1$) violate the transitivity conditions more severely. This means that the clustering of the three objects involved in $t_1$ will increase the value of the total clustering cost $\Delta_{(\boldsymbol{V}, \sigma)}$ more than $t_2$ (also see proof of Theorem 1 in the Appendix for more on this). Therefore, we want to prioritize resolving the inconsistency of the triangle $t_1$ as it will potentially reduce the clustering cost more in the following iteration of the active learning procedure (i.e., when we run the clustering algorithm on the updated similarity matrix). We have added some discussion about this in the second to last paragraph of Section 3.2.

---

### Review · Reviewer_xX2D · 2023-11-26

**Summary Of Contributions:**

This paper considers correlation clustering, where active learning is used to incorporate the setting in which the information about pairwise similarities is given via queries to the oracle. The authors address four limitations of the current approaches: (1) using continuous pairwise similarities instead of binary pairwise similarities, (2) constructing a generic approach, (3) taking noisy oracles into account, and (4) automatically selecting the number of clusters. Empirical evaluation shows superior performance compared to existing active correlation-based or distance-based clustering methods in particular if the proportion of queried pairs is small.

**Audience:**

Yes

**Broader Impact Concerns:**

I do not have any concerns.

**Claims And Evidence:**

No

**Requested Changes:**

Please address all the weakness issues listed above.
In addition, I have the following issues:

- The last sentence in Section 2.2 and the first sentence in Section 3.3 are almost the same. One of them should be removed.
- In Eq.(7), since γ has been already used as the noise level, this γ should be replaced with another greek letter.
- The authors say "the oracle by setting σi+1(u,v) equal to the average of all queries made for the edge (u,v) so far", while I think taking the mode instead of the average could be also effective. Is such an approach considered?

**Strengths And Weaknesses:**

**Strengths**
- An algorithm of active correlation clustering is carefully designed and analyzed. Related work is well discussed.
- Various types of queries have been considered and their effectiveness has been empirically examined, which can be valuable findings for TMLR's audience.
- Empirical performance of the proposed query strategies, in particular **maxmin** and **maxexp**, is promising.

**Weaknesses**
- Although the paper assumes that each query can be noisy; the oracle may return a false answer with probability $1 - \gamma$, this setting is not explicitly considered in the formulation of the proposed method. Moreover, the cost of each query is not considered at all in the formulation, although it is said as "must be queried in a cost-efficient way" in Abstract. Thus, a naïve approach where one makes the same query many times and taking the mode of them, would perfectly work if $\gamma$ is not too small. Thus investigation of the robustness of the proposed method to the noisy oracles is not sufficient.
- Although the computational complexity of the proposed approaches has been analyzed, which itself is a good contribution, the actual running time can vary due to the techniques implemented in the proposal. Therefore it is important to examine and compare the actual running time of proposed query strategies.
- About the time complexity of **maxmin** and **maxexp**, although I understand that it can be efficiently computed in practice using the proposed approach, the worst case still remains to be cubic w.r.t. $N$, and it finally reduces to be quadratic w.r.t. $N$ by random sampling for $\mathbf{E}_{i}^{\text{violates}}$. I think random sampling could affect the resulting performance, hence **maxmin** and **maxexp** without random sampling should be empirically compared.
- Presentation should be improved (please see the requested changes below).

---

> ### Author Response · Authors · 2023-12-07
> **Response to reviewer xX2D**
>
> Thank you for the review and the valuable comments/suggestions! We have attempted to address them in the revised version and in the responses below.
>
> "**Although the paper assumes that each query can be noisy; the oracle may return a false answer with probability $1-\gamma$, this setting is not explicitly considered in the formulation of the proposed method. Moreover, the cost of each query is not considered at all in the formulation, although it is said as "must be queried in a cost-efficient way" in Abstract. Thus, a naïve approach where one makes the same query many times and taking the mode of them, would perfectly work if $\gamma$ is not too small. Thus investigation of the robustness of the proposed method to the noisy oracles is not sufficient."**
>
> **Querying cost:** We assume that the cost of each query (to different pairs) is equal (e.g., 1), though our framework is generic enough to encompass a varying query cost as well (where the acquisition function is divided by the respective cost). This choice means that the problem reduces to minimizing the number of queries made. This is the setting studied in most of the prior active learning work. We have added a comment about this in Section 2.2.
>
> **Noise model:** As described in Section 2.2, we use a non-persistent noise model where the oracle returns the correct pairwise similarity with probability $1-\gamma$, and a noisy value otherwise. Our framework is generic and can admit any arbitrary noise model. The noise model we use in experiments is just one way (but a natural way) to assume a noisy oracle.
>
> As suggested by the reviewer, a naïve approach would be to simply query all pairs sufficiently many times and take the average. However, this would be extremely expensive and will not even be competitive with uniform sampling. The goal is to minimize the number of queries made. The proposed query strategies (maxmin/maxexp) may query the same pair more than once to correct mistakes, but they do so only for a very small subset of the pairs. The power of maxmin/maxexp is therefore in their ability to detect which of the pairs should be queried more than once (instead of naïvely querying all of them multiple times). Because of this, maxmin/maxexp are able to achieve perfect clustering with a querying budget of only 10-30% of total number of edges (i.e., significantly less than querying each edge weight only once). Finally, note that none of our proposed query strategies assumes knowledge of the noise level $\gamma$, as knowing the true value of $\gamma$ would be unrealistic in practice.
>
> **"Although the computational complexity of the proposed approaches has been analyzed, which itself is a good contribution, the actual running time can vary due to the techniques implemented in the proposal. Therefore it is important to examine and compare the actual running time of proposed query strategies."**
>
> We have added Section 4.4 to the revised version, which studies the runtime/performance of maxexp and maxmin. We also would like to add that in active learning, usually the key objective is minimizing the total number of queries (or the respective querying cost) than the computational aspects. Therefore, in this work, we mainly focus on the querying cost.
>
> **"About the time complexity of maxmin and maxexp, although I understand that it can be efficiently computed in practice using the proposed approach, the worst case still remains to be cubic w.r.t. $N$, and it finally reduces to be quadratic w.r.t. $N$ by random sampling for $\mathbf{E}_i^{violates}$. I think random sampling could affect the resulting performance, hence maxmin and maxexp without random sampling should be empirically compared."**
>
> We have added Section 4.4 to the revised version, which studies the runtime/performance of maxexp and maxmin. In particular, we analyze what happens when we sample subsets of $\mathbf{E}_i^{violates}$ of different sizes.
>
> **"The last sentence in Section 2.2 and the first sentence in Section 3.3 are almost the same. One of them should be removed."**
>
> This has been fixed in the revised version.
>
> **"In Eq.(7), since γ has been already used as the noise level, this γ should be replaced with another greek letter."**
>
> We have now updated this by using $\mu$ instead of $\gamma$.
>
> **"The authors say "the oracle by setting $\sigma_{i+1}(u,v)$ equal to the average of all queries made for the edge (u,v) so far", while I think taking the mode instead of the average could be also effective. Is such an approach considered?"**
>
> Using the mode (or other statistics) instead of the average is interesting, and we will investigate this in future work. However, we suspect that the performance of the two methods will be similar.

---

### Review · Reviewer_L3E6 · 2023-12-04

**Summary Of Contributions:**

The authors investigate the problem of correlation clustering and propose a framework to allow active learning under arbitrary correlation clustering algorithms and query strategies. In addition they allow information to be affected by noise and propose to constantly re-estimate the correlation information in an efficient way.

**Audience:**

Yes

**Broader Impact Concerns:**

No concerns.

**Claims And Evidence:**

Yes

**Requested Changes:**

There are two additional experimental conditions that would perhaps be of interest to analyse:

1. in the Experimental Setup section at page 13, the authors describe their initialisation procedure which consists in estimating an initial clustering partition and then setting $\sigma_0$ with +0.1 if two instances are in the same cluster and -0.1 otherwise; however it would be of interest to analyse the performance efficiency when some notion of confidence is allowed, i.e. when a noisy correlation score can be used in the initialisation phase; it is conceivable that users can in fact express an informed opinion on their beliefs. A correct estimate of the correlations/scores should significantly improve the efficiency of the queries and it would be interesting to devise an experimental setup that could show much better efficiencies in this conditions.

2. the initial pairs in $\sigma_0$ are chosen at random. What would happen if instead one could assume some other (more informative) strategy? For example starting from a (noisy) minimum spanning tree? What one should be able to see is that the exploitation of transitivity should be able to yield far greater efficiency than having to query 10-30% of all possible $n^2$ pairs of scores.

**Strengths And Weaknesses:**

The proposal is of interest, it is well written and has a convincing experimental section that shows a clear advantage of the proposed method.
The empirical results demonstrate a clear advantage w.r.t. other approaches, however the setup is such (i.e. no access to the features of the instances but only to their correlations) that the effective number of pairwise similarities that one needs to query is still large in practice (10-30% of all the pairwise similarities in a given dataset). One wonders if by relaxing some assumption greater efficiencies could be obtained (see next part).

---

> ### Author Response · Authors · 2023-12-07
> **Response to reviewer L3E6**
>
> Thank you for the helpful review! We have attempted to address the comments/suggestions in the responses below.
>
> **"The empirical results demonstrate a clear advantage w.r.t. other approaches, however the setup is such (i.e. no access to the features of the instances but only to their correlations) that the effective number of pairwise similarities that one needs to query is still large in practice (10-30% of all the pairwise similarities in a given dataset). One wonders if by relaxing some assumption greater efficiencies could be obtained (see next part)."**
>
> One motivation for assuming no access to feature vectors is that we directly avoid possible noise/ambiguities that may exist in feature space. Additionally, in certain practical applications, one may not even have access to feature vectors because they may be costly to obtain. Instead, in this work, we rely solely on the information coming from the oracle in the form of pairwise similarities. In addition, we admit a noisy oracle. Both of these facts will naturally require more queries to the oracle in order to recover the ground-truth clustering. It should be noted that for applications where a good feature space is available, one can incorporate them as prior information into $\sigma_0$ (as the reviewer is suggesting), which may lead to performance improvement. See response below for more on this.
>
> **"In the Experimental Setup section at page 13, the authors describe their initialisation procedure which consists in estimating an initial clustering partition and then setting $\sigma_0$ with +0.1 if two instances are in the same cluster and -0.1 otherwise; however it would be of interest to analyse the performance efficiency when some notion of confidence is allowed, i.e. when a noisy correlation score can be used in the initialisation phase; it is conceivable that users can in fact express an informed opinion on their beliefs. A correct estimate of the correlations/scores should significantly improve the efficiency of the queries and it would be interesting to devise an experimental setup that could show much better efficiencies in this conditions."**
>
> Please note that we study two different initialization methods: i) we initialize $\sigma_0$ based on a random initial clustering solution, and ii) we use an initial clustering partition based on $k$-means (which does use the feature vectors). The purpose of including the $k$-means initialization is to illustrate the possibility of including prior information in $\sigma_0$ which expectedly does improve the performance. As the reviewer suggests, there may be several other ways to incorporate such prior information that might lead to even better performance. For example, one could initialize $\sigma_0$ proportional to the inverse distance in some feature space. If the feature space is well-defined and is relevant to the true clustering, such an initialization will likely lead to a significant improvement in performance. However, in this paper we mainly focus on the setting where no prior information is available, but in future work we will investigate different ways of initializing $\sigma_0$ and using the features.
>
> "**The initial pairs in $\sigma_0$ are chosen at random. What would happen if instead one could assume some other (more informative) strategy? For example starting from a (noisy) minimum spanning tree? What one should be able to see is that the exploitation of transitivity should be able to yield far greater efficiency than having to query 10-30% of all possible $n^2$ pairs of scores.**"
>
> As described in the answer to the previous comment, in addition to random initialization, we have also investigated initialization based on $k$-means, where the results demonstrate improved performance.

---

### Decision · Action_Editor_VA9j · 2024-02-05

**Recommendation:** Accept as is

**Comment:**

Three reviewers recommended acceptance. During the discussion period, several concerns were raised including relaxing the setup of having no access to the features of the instance, providing more in-depth investigation of the robustness of the proposed framework to the noisy oracles, and providing running time of the proposed query strategies. The revised manuscript adequately addresses these concerns. It's worth noting that the paper focuses on pool-based active learning rather than stream-based active learning, where new data arrives over time, potentially altering its distribution.

**Audience:**

Yes.

**Claims And Evidence:**

The paper introduces an active correlation clustering framework that facilitates active learning with various correlation clustering algorithms and query strategies/acquisition functions. The framework handles a noisy oracle by constantly re-estimating the correlation information in an efficient way. The effectiveness of the approach is thoroughly assessed across ten datasets using five distinct query strategies, and it is compared against three other active correlation clustering approaches.